# Motor cortex can directly drive the globus pallidus neurons in a projection neuron type-dependent manner in the rat

Fuyuki Karube[1]*, Susumu Takahashi[1,2][†], Kenta Kobayashi[3], Fumino Fujiyama[1]*

[1]Laboratory of Neural Circuitry, Graduate School of Brain Science, Doshisha University, Kyotanabe, Japan; [2]Laboratory of Cognitive and Behavioral Neuroscience, Graduate School of Brain Science, Doshisha University, Kyotanabe, Japan; [3]Section of Viral Vector Development, National Institute for Physiological Sciences, Okazaki, Japan

**Abstract** The basal ganglia are critical for the control of motor behaviors and for reinforcement learning. Here, we demonstrate in rats that primary and secondary motor areas (M1 and M2) make functional synaptic connections in the globus pallidus (GP), not usually thought of as an input site of the basal ganglia. Morphological observation revealed that the density of axonal boutons from motor cortices in the GP was 47% and 78% of that in the subthalamic nucleus (STN) from M1 and M2, respectively. Cortical excitation of GP neurons was comparable to that of STN neurons in slice preparations. FoxP2-expressing arkypallidal neurons were preferentially innervated by the motor cortex. The connection probability of cortico-pallidal innervation was higher for M2 than M1. These results suggest that cortico-pallidal innervation is an additional excitatory input to the basal ganglia, and that it can affect behaviors via the cortex-basal ganglia-thalamus motor loop.
DOI: https://doi.org/10.7554/eLife.49511.001

*For correspondence:
fkarube@mail.doshisha.ac.jp (FK);
ffujiyam@mail.doshisha.ac.jp (FF)

Present address: [†]Laboratory of Cognitive and Behavioral Neuroscience, Graduate School of Brain Science, Doshisha University, Kyotanabe, Japan

Competing interests: The authors declare that no competing interests exist.

## Introduction

Parallel loops of neural connections among the cerebral cortex, basal ganglia, and thalamus contribute to multiple aspects of behavior (*Alexander et al., 1986*; *Nambu, 2008*; *Wei and Wang, 2016*). The functions mediated by these loops depend on relevant cortical areas and brain regions receiving outputs of the basal ganglia (*Hikosaka, 2007*; *Middleton and Strick, 2000*). The loop containing the motor cortex is crucial for appropriate motor control, action selection, and movement-related learning. Dysfunction of the motor loop leads to movement disorders such as those seen in Parkinsonian disease (*Albin et al., 1989*; *DeLong, 1990*; *Middleton and Strick, 2002*; *Nambu, 2008*; *Nambu et al., 2000*; *Parent and Hazrati, 1995a*; *Redgrave et al., 2010*; *Wichmann and DeLong, 1996*). Cortical projections drive three pathways in the basal ganglia: the direct, the indirect, and the hyperdirect pathways (*Bolam et al., 2000*; *Smith et al., 1998*). The direct and indirect pathways originate from two distinct types of striatal medium spiny neurons (MSNs), termed the direct- and indirect-pathway MSNs (dMSNs and iMSNs). dMSNs project to the output nuclei of the basal ganglia, namely the substantia nigra (SN) pars reticulata (SNr), and the globus pallidus internal segment, the latter being the homologue of the entopeduncular nucleus (EP) in rodents. iMSNs project to the globus pallidus external segment (GP in rodents), which interconnects with the subthalamic nucleus (STN). In turn, both the STN and GP also innervate the output nuclei. The hyperdirect pathway involves direct cortical projections to the STN (*Nambu et al., 2002*), which provides the fastest information flow among the three pathways (*Nambu et al., 2000*). The behavioral functions of these pathways are gradually being elucidated, although recent findings propose refinement and reappraisal of the classical views of the functional roles of distinct MSNs (*Calabresi et al., 2014*;

*Cui et al., 2013*; *Isomura et al., 2013*; *Vicente et al., 2016*). According to the traditional model, the direct pathway promotes the execution of desired actions, whereas the indirect pathway prevents the execution of competing actions (*Friend and Kravitz, 2014*; *Nambu, 2007*; *Vicente et al., 2016*), and the hyperdirect pathway emergently cancels or switches imminent movements (*Frank et al., 2007*; *Isoda and Hikosaka, 2008*; *Nambu et al., 2002*; *Schmidt et al., 2013*). Massive connections between the STN and GP illustrate that the hyperdirect and the indirect pathways operate in a coordinated manner.

Recently, two types of GP neurons have been identified: one projects to the STN with or without projection to the striatum, while the other only projects to the striatum (*Abdi et al., 2015*; *Dodson et al., 2015*; *Hernández et al., 2015*; *Mallet et al., 2012*; *Mastro et al., 2014*), which has been implicated in the rewriting of neural circuitry involved in the GP. The GP is now considered the hub of the basal ganglia, not simply a relay nucleus (*Hegeman et al., 2016*). Not only the circuitry, but also the functional significance of these neuron types and the entire GP have gradually been uncovered (*Mallet et al., 2016*; *Mastro et al., 2017*); however, how their activities are controlled remains unclear. In 1994, Naito and Kita reported cortical projections to the GP, which raised the possibility of an additional neural pathway that can modulate the basal ganglia activities. Although some studies have proposed more recent evidence on cortico-pallidal projection (*Hunt et al., 2018*; *Abecassis et al., 2019*; *Magno et al., 2019*), a detailed morphological and electrophysiological analysis of this connection has not been conducted so far (*Milardi et al., 2015*; *Smith and Wichmann, 2015*).

Using a combination of neural tracing with immunostaining, we here demonstrate that M1 and M2 directly project to the GP, and that cortical axon collaterals and boutons have topographic distributions that depend on the cortical area of origin (*Naito and Kita, 1994*). Using morphological and electrophysiological experiments combined with optogenetics, we demonstrate that cortico-pallidal synapses are effective and dependent on post-synaptic GP cell types. We discuss the potential roles of cortico-pallidal projections in relation to other basal ganglia nuclei. Part of this research has been presented in abstract form earlier (*Karube and Fujiyama, 2017*).

## Results

### Motor cortex innervates the GP

We observed cortical axons in the GP using conventional tracers (BDA or PHA-L; *Figure 1A, B and C*; *Figure 2*, *Figure 1—figure supplements 1*, *2* and *3*; *Figure 2—figure supplement 1*) or adenoassociated virus (AAV) vectors (*Figure 1D*; *Figure 2—figure supplements 1*, *2*), as reported previously (*Naito and Kita, 1994*). For both M1 and M2 projections, cortical axons were dense in calbindin D-28k (CB) -negative [CB(-)] regions of the GP. The appearance of cortical axon varicosities in the GP was similar to those in the striatum and STN (*Figure 1A2 and C*, *Figure 1—figure supplements 1* and *2*). Some axons in the GP were emitted from thick trunks, and often elongated along the dorso-ventral axis (*Figure 1B*). Cortico-pallidal projections were exclusively ipsilateral (*Figure 1D*), implying that they did not stem from the intratelencephalon (IT) type of pyramidal neurons. In addition, we found that cingulate area (Cg) and lateral orbitofrontal area (LO) also projected to the GP (*Figure 2—figure supplements 1* and *2*), although LO provided fewer axons than motor areas. Cortical axons were differentially distributed in the GP, depending on the cortical area of origin (*Figure 1—figure supplements 1* and *2*; *Figure 2—figure supplements 1* and *2*). M1 and M2 axons were observed across the broad extent of the medio-lateral axis (range: M/L 2.4–4.2 mm from the midline; see also *Figure 2A, B and C*), but relatively sparse in the medial and lateral portions in most cases (*Figure 2C*). LO axons were often located in the ventral part of the GP, whereas Cg axons were dense in the more medial GP than M1/M2 axons.

We compared the density of cortical axonal varicosities (boutons) in the GP, STN, and striatum. The number of varicosities in the GP varied with the M/L dimension, as in the case of varicosities in the STN for both M1 and M2 projections (*Figure 2C*; $N = 4$ rats for each). Varicosity density was apparently greater in the striatum than in the GP or STN (*Figures 1A* and *2B*, *Figure 1—figure supplements 1* and *2*, *Table 1*). Since the size and efficacy of the tracer labeling were not uniform across animals, the maximum bouton density in the GP was normalized to that in the striatum or STN for each injection. Using this normalization, the bouton density was found to be significantly

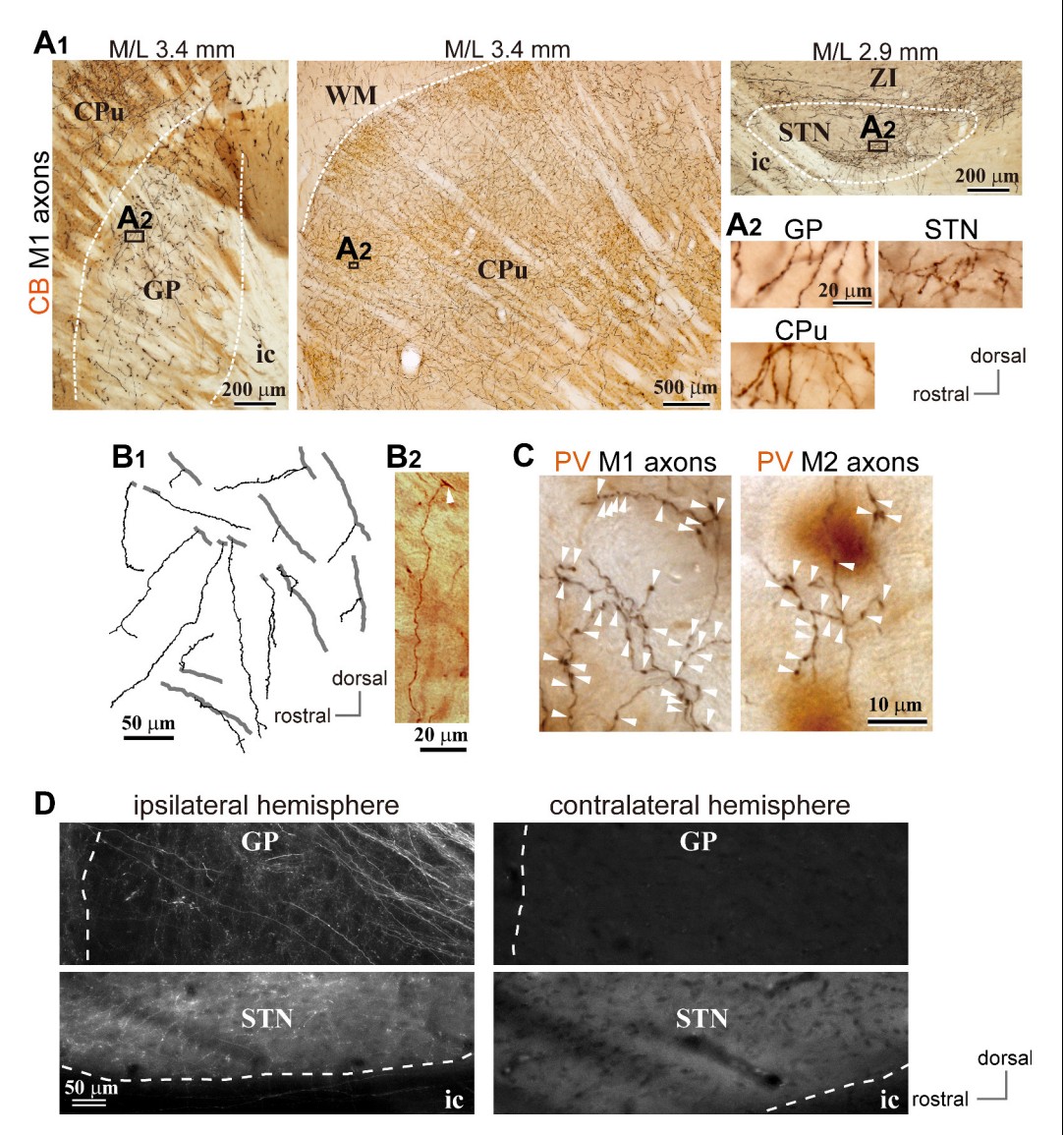

**Figure 1.** Motor cortical axons project to the globus pallidus (GP). (**A**) Representative images of axons in the GP, striatum, and STN originated from the primary motor cortex (M1), labeled with biotinylated dextran amine (BDA; visualized in black). Cortical axons in the GP are predominantly distributed in the calbindin (CB)-negative regions. CB is visualized in brown. Irrespective of differences in axon density among the GP, striatum, and STN, the morphology of the axons and varicosities is similar (A2). (**B**) Drawings and an image of cortical axons in the GP. The collaterals (thin black lines in B1) were issued from thick axon trunks (thick gray lines in B1; arrowhead in B2). (**C**) Magnified views of cortical axon varicosities (arrowheads) in the GP; left, M1 axons; right, M2 axons. The images are composites from multiple focal planes. The sections were counter-stained with an anti-parvalbumin (PV) antibody, visualized in brown. (**D**), Cortical axons in the GP were found exclusively in the ipsilateral hemisphere. M2 axons were visualized with AAV. Similar to axons in the STN, fluorescent signals were not detected in the contralateral GP even by over-exposure.
DOI: https://doi.org/10.7554/eLife.49511.002

The following figure supplements are available for figure 1:

**Figure supplement 1.** Example images of M1 projections to the GP, STN, and striatum (related to *Figure 1*).
DOI: https://doi.org/10.7554/eLife.49511.003

**Figure supplement 2.** Example images of M2 projections to the GP, striatum, and STN (related to *Figure 1*).
DOI: https://doi.org/10.7554/eLife.49511.004

**Figure supplement 3.** Example images of M2 projections to the striatum and GP (related to *Figure 1*).
DOI: https://doi.org/10.7554/eLife.49511.005

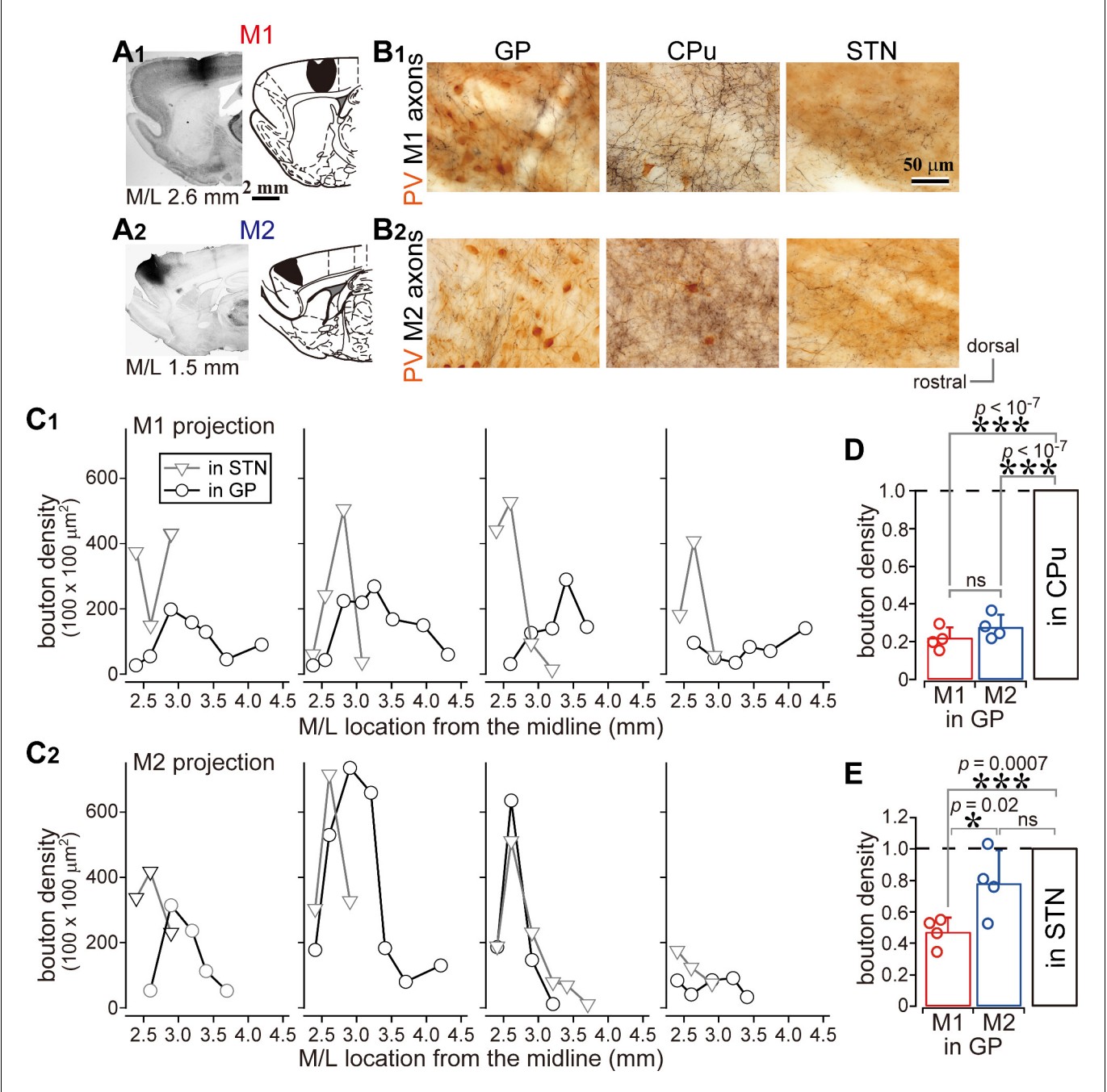

**Figure 2.** Comparison of axonal bouton density in the GP, striatum, and STN. (A), Images and drawings of BDA injection sites into the primary and secondary motor areas (M1 and M2, respectively). Tracer was deposited across the entire thickness of the cortex. (B) Images of axon distributions in the GP, striatum, and STN from M1 and M2. To clearly represent axon density, the images are composites from multiple focal planes. (C) Comparison of axon varicosity density in the GP with that in the STN along the M/L dimension (N = 4 animals for each M1 and M2). In most cases, varicosities in the STN showed a sharp peak at a given M/L position. In the GP, M1 axon varicosities were broadly distributed along the M/L dimension. M2 axons tended to be densely distributed in the medial portion. (D) The axon varicosity density in the GP was normalized to that in the striatum or the STN (E).
DOI: https://doi.org/10.7554/eLife.49511.006

The following source data and figure supplements are available for figure 2:

**Source data 1.** Source data for *Figure 2C*.
DOI: https://doi.org/10.7554/eLife.49511.009

**Source data 2.** Source data for *Figure 2D and E*, *Figure 1—figure supplement 2E*, and *Figure 2—figure supplement 1D*.
DOI: https://doi.org/10.7554/eLife.49511.010

*Figure 2 continued on next page*

*Figure 2 continued*

**Figure supplement 1.** LO projections to the basal ganglia (related to *Figure 2*).
DOI: https://doi.org/10.7554/eLife.49511.007
**Figure supplement 2.** Cingulate cortex (Cg) projections to the basal ganglia (related to *Figure 2*).
DOI: https://doi.org/10.7554/eLife.49511.008

higher in the striatum than in the GP (*Figure 2D*; p=0.013, Kruskal Wallis test; p<10$^{-7}$ for both M1 and M2 projections, Tukey-HSD test; not significant between M1 and M2, p=0.18), or in the STN (*Figure 1—figure supplement 2E*; p=0.016, Kruskal Wallis test; p=0.00003 for M1; p=0.000008 for M2, Tukey-HSD test; not significant between M1 and M2, p=0.296). Normalized bouton densities in the GP were significantly different from those in the STN (*Figure 2E*; p=0.032, Kruskal-Wallis test). For M1 projections, STN-normalized bouton density was significantly smaller in the GP (0.496 ± 0.12) than in the STN (p=0.0007, Tukey-HSD test). Since M2 had more boutons in the GP (0.778 ± 0.207)

**Table 1.** Electrophysiological properties of globus pallidus (GP) neurons.

| | GP$_{STN}$ (*N* = 108) | GP$_{CPu}$ (*N* = 65) | *p*-value |
|---|---|---|---|
| Mean membrane potential (mV) | −46.63 ± 4.95 (−34.2− −61.53) | −46.36 ± 5.9 (−33.59− −60.53) | 0.682 |
| Input resistance (MΩ) | 232.35 ± 131.87 (33.1–887.45) | 329.97 ± 168.19 (32.8–828.3) | 6.8 × 10$^{-05}$ *** |
| Time constant (ms) | 12.94 ± 10.23 (2.25–79.75) | 21.92 ± 13.11 (2.42–55.67) | 2.1 × 10$^{-07}$ *** |
| Sag potential (mV) | 7.23 ± 4.52 (1.25–22.02) | 8.71 ± 7 (0.63–35.18) | 0.3432 |
| Spike frequency at 100 pA depolarization (Hz) | 48.46 ± 22.94 (0–103) | 35.81 ± 19.6 (2–85) | 0.00041 *** |
| Spike frequency at 500 pA depolarization (Hz) | 99.14 ± 66.87 (1–244) | 35.24 ± 39.24 (1–172) | 1.8 × 10$^{-09}$ *** |
| Maximum spike frequency (Hz) | 135.04 ± 65.42 (11–263) | 72.14 ± 36.12 (2–172) | 1.0 × 10$^{-09}$ *** |
| Current at maximum spike frequency (pA) | 578.63 ± 277.97 (50–1000) | 345.24 ± 219 (50–1000) | 6.9 × 10$^{-08}$ *** |
| Spike height (mV) | 74.06 ± 9.77 (47.04–96.98) | 76.55 ± 11.69 (38.84–95.97) | 0.051 |
| Spike width (ms) | 0.95 ± 0.29 (0.63–2.29) | 1.19 ± 0.32 0.7–2.17 | 1.6 × 10$^{-09}$ *** |
| Spike threshold (mV) | −37.57 ± 4.79 (−24.09− −48.11) | −37.7 ± 5.32 (−23.14− −46.03) | 0.5351 |
| fAHP amplitude (mV) | 21.16 ± 5.14 (9.1–38.72) | 18.09 ± 4.57 (11.88–36.34) | 1.3 × 10$^{-05}$ *** |
| fAHP delay after spike peak (ms) | 0.98 ± 0.39 (0.55–2.7) | 1.31 ± 0.55 (0.6–3.8) | 3.4 × 10$^{-07}$*** |
| sAHP amplitude (mV) | 18.26 ± 4.2 (9.47–32.87) | 16.54 ± 4.71 (10.38–29.78) | 0.0035 ** |
| sAHP delay after spike peak (ms) | 9.63 ± 3.32 (2.25–16.95) | 22.58 ± 13.54 (2.25–65.5) | 4.5 × 10$^{-12}$ *** |
| On cell mode spontaneous firing frequency (Hz) | 20.79 ± 18.74 (0–90.53) | 5.47 ± 9.22 (0–35.68) | 1.2 × 10$^{-08}$ *** |
| Whole cell mode spontaneous firing frequency (Hz) | 19.02 ± 13.74 (0–61.74) | 9.41 ± 9.44 (0–31.85) | 4.0 × 10$^{-06}$ *** |

GP$_{STN}$, GP neurons projecting to the subthalamic nucleus; GP$_{CPu}$, GP neurons projecting to the striatum; fAHP, fast afterhyperpolarization; sAHP, slow afterhyperpolarization; **, p<0.01; ***, p<0.001. Statistical significance was examined using the Wilcoxon rank sum test. The range of each parameter is shown in parentheses.
DOI: https://doi.org/10.7554/eLife.49511.012

than M1 (p=0.022), M2 bouton density did not significantly differ between the STN and the GP (p=0.09).

These findings suggest that the motor cortices provide a substantial number of boutons in the GP, and can thus modulate GP neuronal activity as they affect the striatum and the STN. However, because bouton density does not directly indicate synaptic efficacy, we also evaluated the electrophysiological features of cortico-pallidal projections.

### Cortico-GP terminals elicit monosynaptic EPSCs

We conducted whole-cell patch clamp recordings combined with optogenetic stimulation of cortical terminals using in vitro slice preparations. Channelrhodopsin 2 (ChR2) was introduced into cortical neurons using *AAV-hSyn-H134R-mCherry* injections into either M1 or M2 (*Figure 3A*). In voltage clamp mode at a holding potential of −60 mV, stimulation with a brief light pulse (5 ms, 470 nm) elicited inward currents in GP neurons (*Figure 3B1*). The response was stable over repetitive stimulation (10 pulses at 2–10 Hz; *Figure 3B1*). In current clamp mode, photoactivation elicited action potentials, although the action potential probability was affected by the spontaneous oscillation of the membrane potential (*Figure 3B2*). To confirm that the photoactivated current that elicited action potentials was within the physiological range, we measured the minimum current required to induce

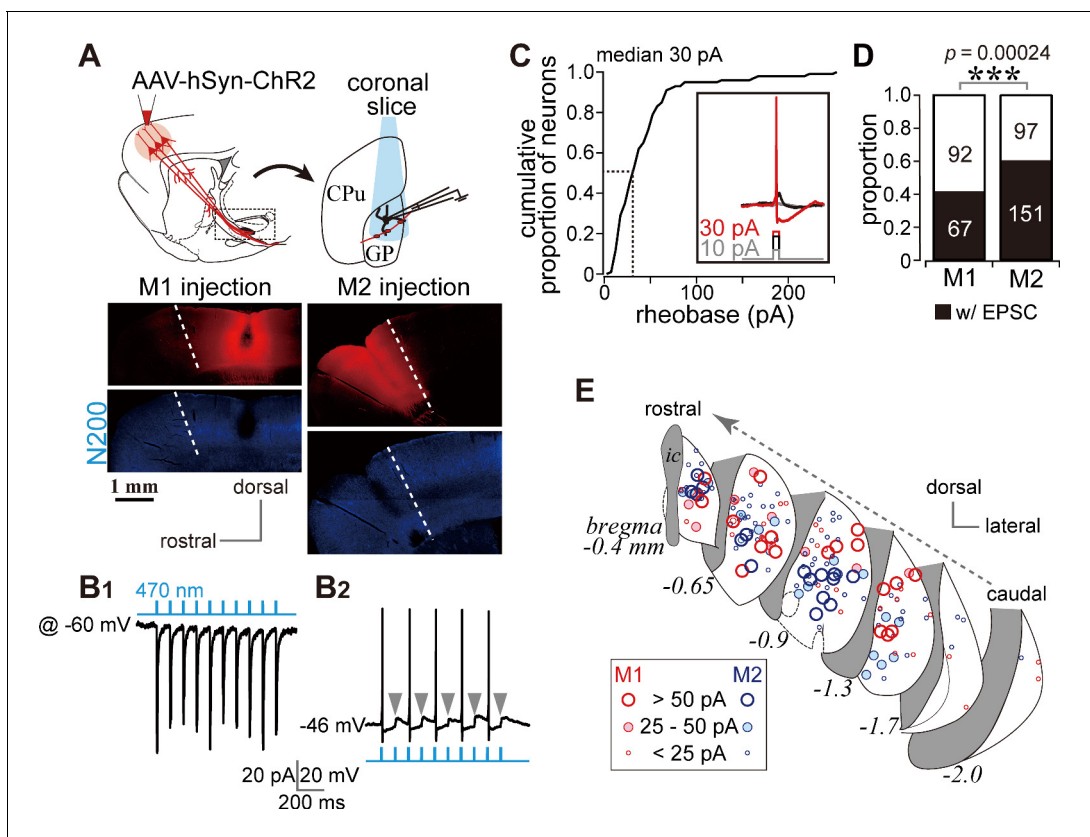

**Figure 3.** Photoactivation of motor cortical terminals evokes excitation in GP neurons. (**A**) Schematic (top) of AAV encoding channelrhodopsin 2 and mCherry injection into the motor cortex for ex vivo recordings using coronal slices. Examples of AAV injection sites are shown in the middle panels (red). Images of immunofluorescence for neurofilament 200 kDa (N200, bottom), used for identification of the M1/M2 border (white dotted lines). (B1) A representative voltage clamp trace (held at −60 mV) showing inward currents in GP neurons elicited by 5 ms blue light pulses (470 nm). (B2) A representative current clamp trace showing photoinduced action potentials and excitatory postsynaptic potentials (EPSPs, arrowheads). (**C**) Cumulative histogram of the rheobase current of GP neurons. Note that 25 to 30 pA is sufficient to elicit action potentials in half of GP neurons (N = 100). (**D**) Proportion of GP neurons innervated by M1 or M2 terminals. The number of neurons is shown in bars. M2 more frequently innervated the GP than did M1. (**E**) Location of GP neurons innervated by M1 (red circle) or M2 (blue circle). Note the topographic distribution of M1 and M2 innervation. The size of circles represents the amplitude of optically evoked currents.
DOI: https://doi.org/10.7554/eLife.49511.011

action potentials (rheobase current) in GP neurons using 5 ms depolarizing pulses (*Figure 3C*, inset). In half of the GP neurons, the rheobase was less than 30 pA, and most GP neurons could emit an action potential with less than 100 pA of depolarization (*N* = 100 neurons; *Figure 3C*). A depolarized membrane potential and a high input resistance (*Table 1*) led to easy induction of action potentials by small excitation.

Not all GP neurons exhibited inward photocurrents, a total of 67/159 and 151/248 neurons did so during M1 and M2 stimulation, respectively (*Figure 3D*). The locations of the GP neurons in which inward currents were observed were plotted (*Figure 3E*). Consistent with the distribution of cortical axons, these locations were frequently around the center of the GP in coronal slices. Responsive neurons were similarly concentrated around the center of the GP along the rostro-caudal axis. Neurons responding to M1 terminal stimulation tended to be located in the dorsal GP, whereas those responding to M2 terminal stimulation were clustered in the ventral GP (*Figure 3E*).

It is possible that the observed EPSCs were elicited by the STN via a di-synaptic circuit. However, we used coronal slices with an anteroposterior position of 0.6 mm rostral (r0.6)–2.2 mm caudal (c2.2) to bregma, which did not include the STN (*Paxinos and Watson, 2007*). Bath application of the sodium channel blocker tetrodotoxin (TTX) at 1 µM completely prevented inward currents (*Figure 4A*). Additional application of the potassium channel blocker 4-aminopyridine (4AP) at 1 mM recovered the currents to up to 60% of control on average (*Figure 4A*), indicating that the current was monosynaptic (*Gradinaru et al., 2009*; *Shu et al., 2007*). A GABA$_A$ receptor antagonist (gabazine, 20 µM) did not block the current (*Figure 4A*); conversely, glutamate receptor antagonists (CNQX, 10 µM and AP-5, 20 µM) almost completely abolished the current (*Figure 4A*). Thus, the inward current was mediated by glutamatergic excitatory postsynaptic currents (EPSCs). We compared the time course of EPSCs between STN and GP neurons. The latency (delay) of current onset after the photic stimulus in GP neurons did not differ from that in STN neurons (*Figure 4B*), which indicates that both EPSCs are directly elicited by cortical terminal activation. The 20–80% rise time and decay constant tended to be longer in GP neurons than in STN neurons (*Figure 4B*). Taken together, these results indicate that the inward current observed in GP neurons was a monosynaptic, glutamatergic EPSC directly elicited by cortical terminals. Hereinafter, we refer to this current as an optically evoked EPSC (oEPSC). In some experiments, both GP and STN neurons were recorded from the same animal, and the amplitude of oEPSC in GP neurons was normalized to the mean of oEPSCs from STN neurons (*Figure 4C*; *N* = 10 GP neurons from three rats for M1 stimulation and *N* = 13 GP neurons from two rats for M2 stimulation). The normalized mean values were not significantly different from 1 (p>0.1, signed rank test), suggesting that the motor cortex could innervate GP and STN using similar weights, however, as the oEPSC amplitude largely varied across cells in both GP and STN neurons, the interpretation of this comparison must be made with caution.

## M1 and M2 more frequently innervate GP neurons projecting to the striatum than those projecting to the STN

The results shown in *Figure 3D* raised the question of whether the cortical innervation of GP neurons follows any specific rules. To determine whether cortical innervation depends on the GP-neuron projection type, we conducted in vitro whole cell recordings from retrogradely labeled GP$_{STN}$ or GP$_{CPu}$ neurons (*Figure 5A*). Occasionally, we recorded large GP neurons with no or very little spontaneous activity (*N* = 6, 1.20 ± 1.79 Hz during on-cell recording mode), which exhibited a distinct action potential shape. Based on earlier reports (*Bengtson and Osborne, 2000*; *Hernández et al., 2015*), these were most likely cholinergic neurons and were excluded from subsequent analysis, although they were found to be innervated by the cortex (*N* = 5/6). In addition to the molecular profiles, the electrophysiological properties of GP$_{CPu}$ and GP$_{STN}$ neurons also differed. GP$_{STN}$ neurons usually showed spontaneous repetitive firing (~20 Hz; *Table 1*), whereas many GP$_{CPu}$ neurons were silent. Firing frequencies induced by depolarizing current pulses were higher in GP$_{STN}$ than in GP$_{CPu}$ neurons, and spike width was narrower in GP$_{STN}$ neurons (*Figure 5B*; see *Table 1* for other electrophysiological parameters and quantitative comparisons). We discovered that GP$_{CPu}$ neurons (51/62) responded more frequently to opotogenetic activation of motor cortical afferents than GP$_{STN}$ neurons (53/126), as shown in *Figure 5C*. The oEPSC amplitude was also larger in GP$_{CPu}$ neurons (*Figure 5D,E,F*). The distribution of oEPSC amplitudes recorded from GP$_{CPu}$ neurons seemed bimodal; the smaller-amplitude group was similar to GP$_{STN}$ neurons (*Figure 5D*). For comparison, MSNs (*N* = 11) and STN neurons (*N* = 18) were also recorded. One-way ANOVA followed by post-

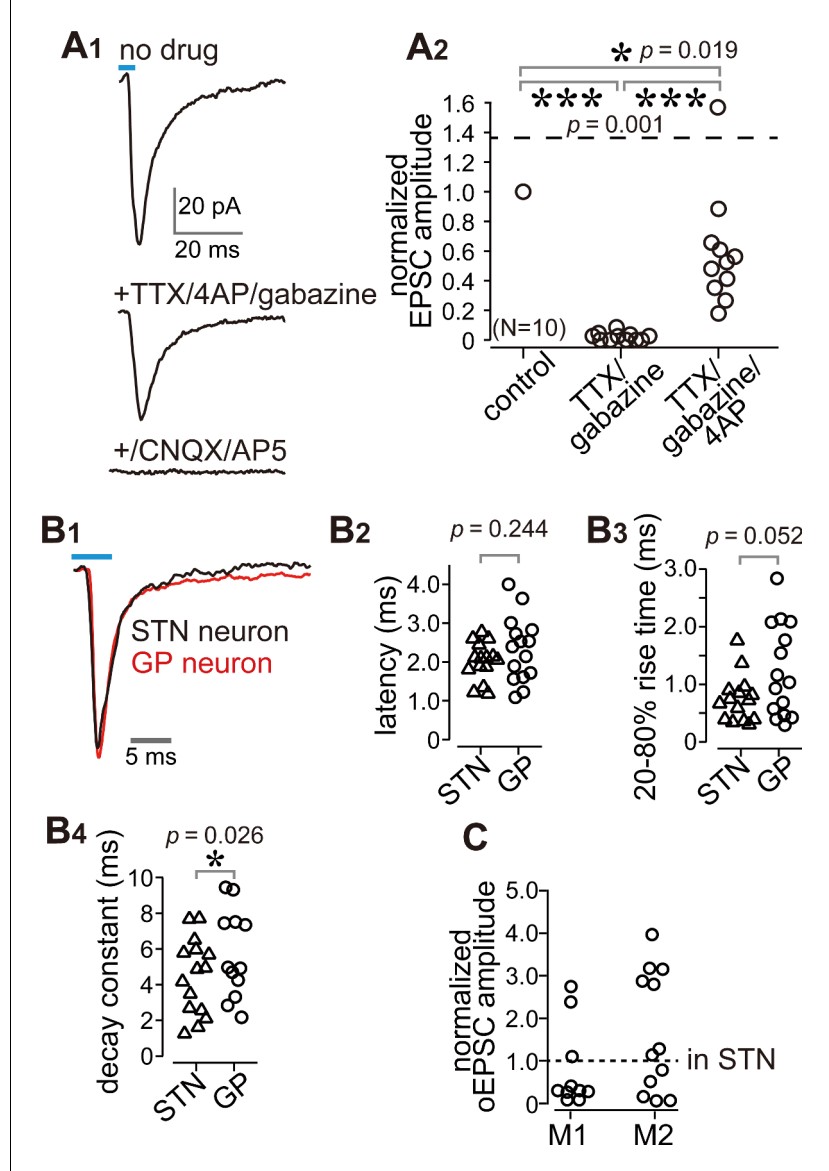

**Figure 4.** Cortico-pallidal connections are monosynaptic and glutamatergic. (**A**) The current induced by optogenetic stimulation of cortical terminals is monosynaptic and glutamatergic. (**A1**) Representative traces of pharmacological effects on the light-induced inward current. Top, no treatment; Middle, effects of TTX, 4AP, and gabazine; Bottom, effects of additional application of glutamate receptor antagonists (CNQX, AP5). (**A2**) A summary plot of pharmacological treatments (*N* = 10 GP neurons). (**B**) Comparison of the time courses of optogenetically evoked EPSCs (oEPSCs). (**B1**) Representative traces recorded from STN (black) and GP (red) neurons. The GP and STN neurons were not recorded simultaneously. (**B2-4**) Summary plots of oEPSC time courses in STN and GP neurons. (**B2**) The latency from light onset to oEPSC onset does not differ between STN and GP neurons (both *N* = 15 neurons). The rise time (**B3**) and the decay constant (**B4**) of oEPSCs tended to be longer in GP neurons (*N* = 15 GP neurons and 12 STN neurons). (**C**), Comparison of oEPSC amplitude between GP and STN neurons, derived from the same animal. The amplitude in the GP neurons was normalized to that in the STN. No significant difference was observed between GP and STN neurons.
DOI: https://doi.org/10.7554/eLife.49511.013

The following source data is available for figure 4:

**Source data 1.** Source data for *Figure 4A*.
DOI: https://doi.org/10.7554/eLife.49511.014
**Source data 2.** Source data for *Figure 4B and C*.
DOI: https://doi.org/10.7554/eLife.49511.015

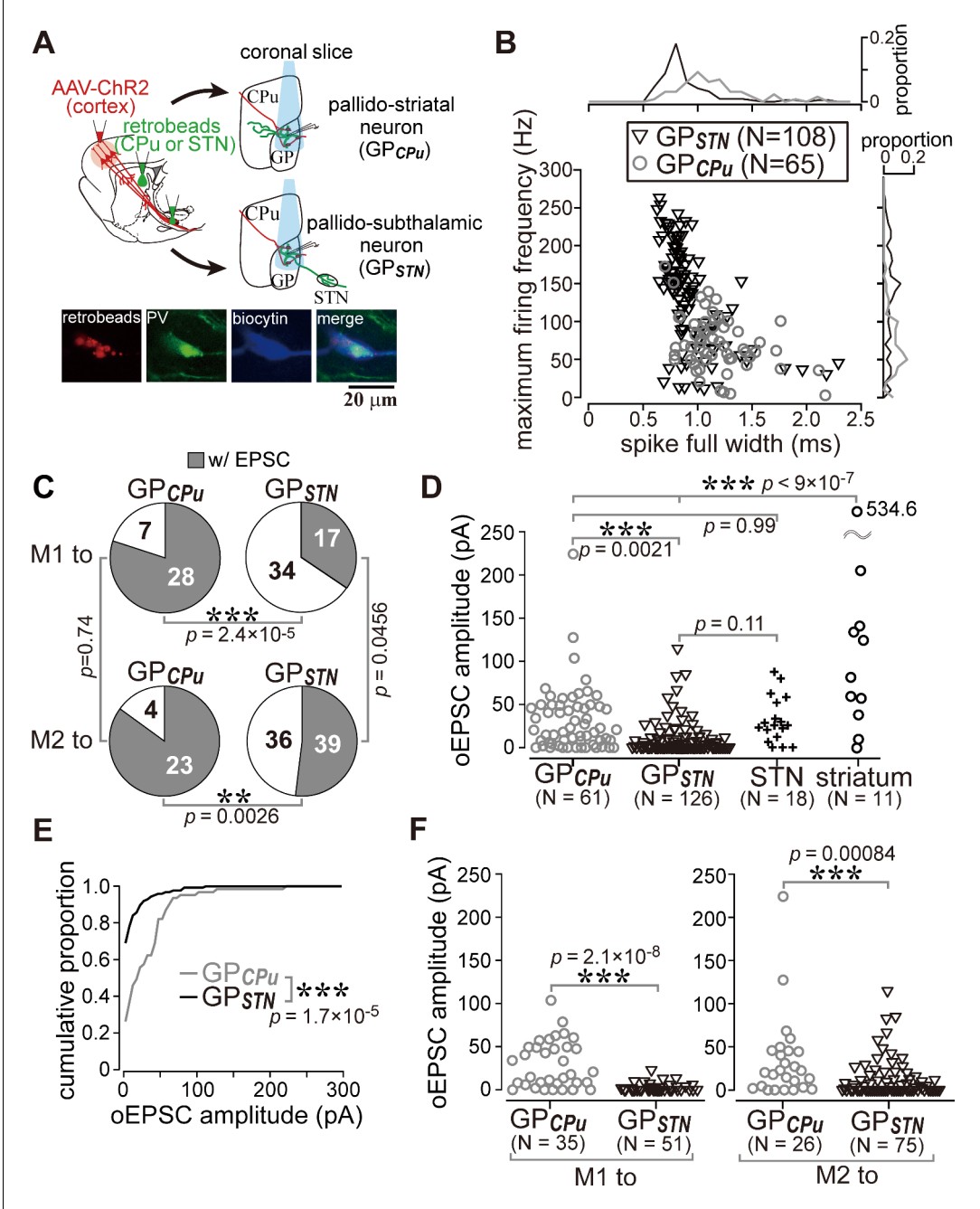

**Figure 5.** Cell type-dependent cortical innervation of pallidal neurons. (**A**) Schematic of ex vivo recordings from retrogradely labeled GP neurons to investigate the effect of GP neuron projection type on cortical innervation. (**B**) Electrophysiological differences between GP neurons projecting to the striatum (GP$_{CPu}$) and STN (GP$_{STN}$) (see also *Table 1*). (**C**) The proportion of GP neurons innervated by M1 or M2 is correlated with projection type. GP$_{CPu}$ neurons were more often innervated by M1 or M2 than GP$_{STN}$ neurons. Significance was examined using Fisher's exact test with a Bonferroni correction for multiple comparisons (***, p<0.00025; **, p<0.0025; *, p<0.0125). (**D**) Amplitudes of oEPSCs in GP, STN, and striatal neurons. The amplitude in GP neurons is similar to that in STN neurons but smaller than that in striatal medium spiny neurons (MSNs). Data obtained from M1 and M2 stimulation are summed. (**E**) Cumulative histograms of oEPSC amplitude in GP$_{CPu}$ and GP$_{STN}$ neurons. GP$_{CPu}$ neurons exhibit a greater optically evoked EPSC (oEPSC) amplitude. (**F**) Left, amplitudes of oEPSCs induced in GP neurons by M1 terminal stimulation. The GP$_{CPu}$ group shows larger oEPSC amplitudes than the GP$_{STN}$ group. Right, amplitudes of oEPSCs induced by M2 terminal stimulation. Significantly larger oEPSCs were again recorded in the GP$_{CPu}$ group (p=0.00084), but the difference is small.

*Figure 5 continued on next page*

*Figure 5 continued*

DOI: https://doi.org/10.7554/eLife.49511.016

The following source data is available for figure 5:

**Source data 1.** Source data for *Figure 5B*.

DOI: https://doi.org/10.7554/eLife.49511.017

**Source data 2.** Source data for *Figures 5C, D, E and F*.

DOI: https://doi.org/10.7554/eLife.49511.018

hoc Tukey tests revealed that the oEPSC amplitude was significantly larger in MSNs than in STN or GP neurons (for all combinations of comparisons, $p < 9 \times 10^{-7}$). The oEPSC amplitude of STN neurons was not significantly different from that of $GP_{STN}$ (p=0.109) or $GP_{CPu}$ neurons (p=0.999). $GP_{CPu}$ neurons exhibited significantly greater amplitudes than $GP_{STN}$ neurons (p=0.0021; *Figure 5D*). We observed relatively hyperpolarized membrane potentials and lower membrane input resistances in MSNs compared with GP or STN neurons. The mean membrane potential ($V_{mean}$) was $-73.45 \pm 5.3$ mV, and input resistance ($R_{in}$) was $79.17 \pm 28.38$ MΩ for 10 MSNs, whereas $V_{mean}$ was $-45.43 \pm 7.47$ mV and $R_{in}$ was $253.14 \pm 156.31$ MΩ for 15 STN neurons (see *Table 1* for GP neurons). Thus, despite the larger EPSCs, the ease of induction of action potentials was lower in MSNs than in GP or STN neurons, at least in slice preparations. Actually, the median of the rheobase current was 695 pA for MSNs ($N = 10$; range, 450–1415 pA), 55 pA for STN neurons ($N = 30$; range, 10–545 pA) and 30 pA for GP neurons ($N = 100$; range, 5–250 pA).

$GP_{CPu}$ neurons were frequently innervated by either M1 (28/35) or M2 (23/27). In contrast, only a small fraction of $GP_{STN}$ neurons received M1 (17/51) or M2 (36/75) inputs (*Figure 5C*). The amplitude of M1-induced oEPSCs was significantly larger in the $GP_{CPu}$ neurons than in the $GP_{STN}$ neurons ($p=2.1 \times 10^{-8}$, Wilcoxon rank sum test). This was also the case for M2-induced oEPSCs (p=0.00084). Therefore, both motor areas preferentially innervated $GP_{CPu}$ neurons, although $GP_{STN}$ was more effectively innervated by M2 (*Figure 5C F*). Indeed, the mean oEPSC amplitude in $GP_{STN}$ was larger with M2 stimulation than with M1 stimulation (p=0.0028), but no significant difference between cortical sites was observed in $GP_{CPu}$ neurons (p=0.9595). M2 stimulation frequently evoked oEPSCs with initially smaller amplitudes, especially for the first light pulse, although repetitive light pulses at 10 Hz often augmented the oEPSC amplitude, similar to M1 stimulation. The paired-pulse ratio for second-to-first oEPSC in $GP_{STN}$ neurons was $1.27 \pm 0.81$ for M2 stimulation ($N = 38$) and $1.45 \pm 1.20$ for M1 stimulation ($N = 14$); in the $GP_{CPu}$ neurons, it was $1.25 \pm 0.62$ ($N = 21$) for M2 stimulation and $1.31 \pm 0.79$ ($N = 28$) for M1 stimulation.

It is important to uncover if any innervation bias exists in $GP_{CPu}$ neurons, because $GP_{CPu}$ neurons should be composed of at least two neuron types, prototypic neurons and arkypallidal neurons. Prototypic neurons are composed of parvalbumin (PV)- and/or LIM homeobox 6 (Lhx6)- expressing neurons, and Lhx6 neurons provide more axons into the striatum than PV neurons (*Hernández et al., 2015*; *Mizutani et al., 2017*; *Oh et al., 2017*). In contrast, arkypallidal neurons express forkhead box protein 2 (FoxP2), and exclusively project to the striatum. Electrophysiological membrane properties are known to differentiate between arkypallidal and prototypic neurons (*Abdi et al., 2015*; *Hernández et al., 2015*; *Mallet et al., 2012*; *Mastro et al., 2014*), however, as shown in *Figure 5B*, the range of parameter values of $GP_{STN}$ and $GP_{CPu}$ neurons overlapped, indicating that they do not allow to distinguish cell types strictly. In addition, we found that immunodetection of Lhx6 was somewhat unstable after in vitro slice recordings.

Since at least a part of the Lhx6 neurons projects to both the striatum and the STN, we hypothesized that double retrograde labeling could distinguish them from other GP neuron types. To confirm this, using retrograde labeling of GP neurons and immunofluorescence against PV, Lhx6, and FoxP2, we examined the molecular profiles of GP neuron types in Wistar rats (*Fujiyama et al., 2016*). We confirmed that $GP_{CPu}$ neurons frequently expressed FoxP2 or Lhx6, but not PV ($N = 659$ $GP_{CPu}$ neurons in nine sections from three rats; *Figure 6A*), in agreement with previous studies in mice (*Dodson et al., 2015*; *Hernández et al., 2015*; *Mastro et al., 2014*; *Mizutani et al., 2017*), Long-Evans rats (*Oh et al., 2017*), and Sprague-Dawley rats (*Abdi et al., 2015*; *Kita and Kita, 2001*). The expression of FoxP2 and Lhx6 was almost mutually exclusive. Most PV(+) $GP_{CPu}$ neurons co-expressed Lhx6. $GP_{STN}$ neurons lacked expression of FoxP2 but expressed PV and/or Lhx6

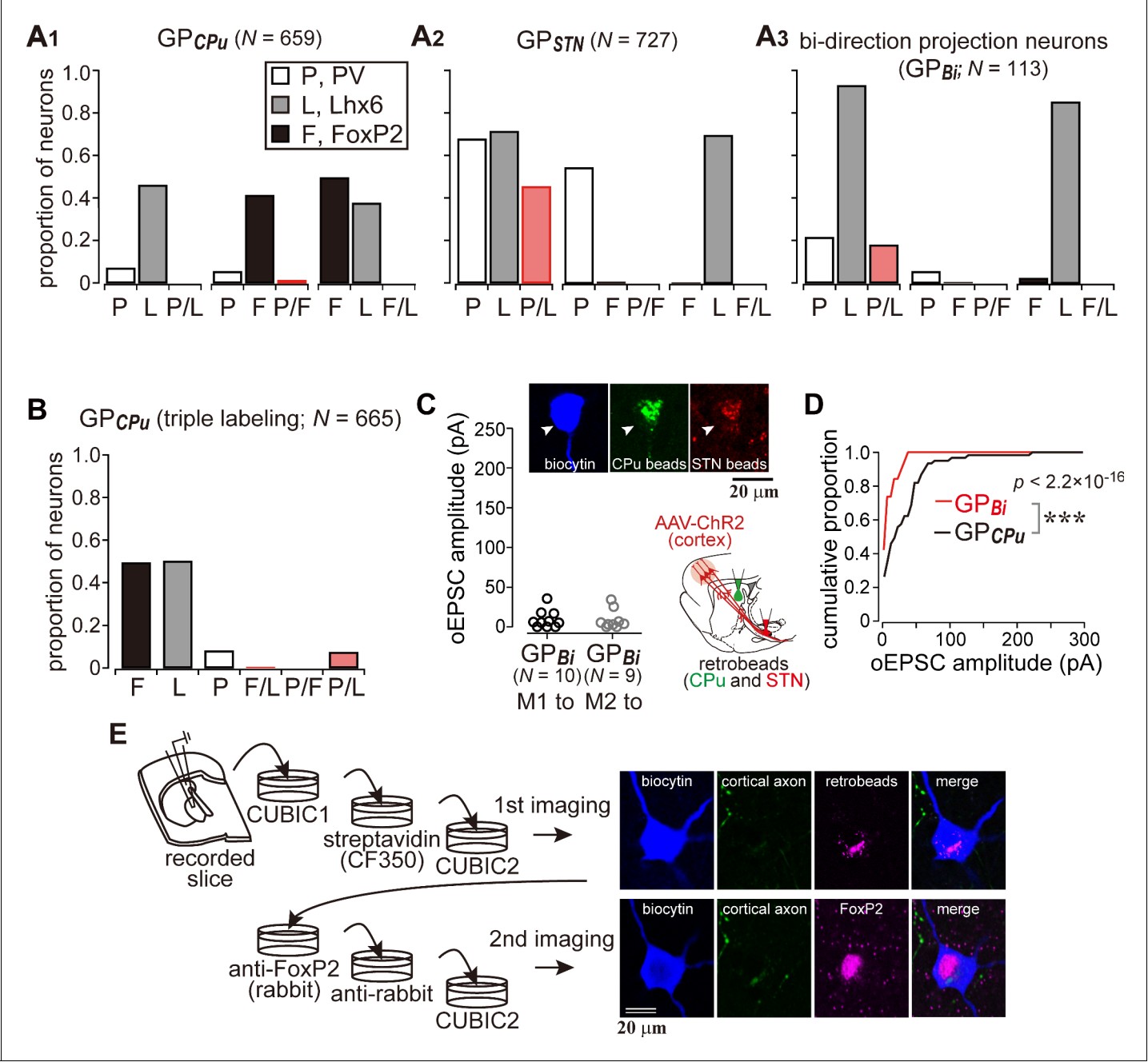

**Figure 6.** Cortical inputs on bi-directional projection GP neurons (GP_Bi) and arkypallidal neurons. Two distinct retrograde tracers were injected into the STN and striatum (CPu), respectively. (**A**) Three combinations of double immunofluorescence (PV/Lhx6, PV/FoxP2, and Lhx6/FoxP2) were applied. CPu-projecting GP neurons (GP_CPu) expressed either FoxP2 or Lhx6 exclusively (A1), whereas STN-projecting GP neurons (GP_STN) were mainly composed of PV- and/or Lhx6-expressing neurons (A2). (A3) GP neurons projecting to both the CPu and STN (GP_Bi) frequently expressed Lhx6, but not FoxP2. (**B**) Triple immunofluorescence for GP_CPu neurons. Only a single retrograde tracer was injected into the striatum. (**C**) oEPSC amplitudes of GP_Bi neurons. Most GP_Bi neurons exhibited small amplitudes of oEPSCs. Green and red retrobeads were injected into the CPu and STN, respectively. Confocal images of a biocytin-filled GP_Bi neuron (arrowheads) are shown. AAV-labeled cortical axons also show red fluorescence. (**D**) Cumulative histograms of oEPSC amplitudes in GP_Bi and GP_CPu neurons. The distributions are significantly different (p=2.2 × $10^{-16}$ Kolmogorov-Smirnov test). The histogram of GP_CPu neurons is the same as that shown in *Figure 5E*. (**E**) An example of confocal images of GP_CPu neurons innervated by the motor cortex. Upper, the first imaging session before immunoreaction to identify a biocytin-filled neuron that is labeled with retrobeads. The slice was cleared with CUBIC. Bottom, the second imaging session of the neuron after immunoreaction against FoxP2. The focal plane was slightly shifted from the first to the second session, because retrobeads are accumulated in the cytosol, whereas FoxP2 is localized in the nucleus.

DOI: https://doi.org/10.7554/eLife.49511.020

*Figure 6 continued on next page*

*Figure 6 continued*

The following source data is available for figure 6:

**Source data 1.** Source data for *Figure 6A and B*.
DOI: https://doi.org/10.7554/eLife.49511.021
**Source data 2.** Source data for *Figure 6C and D*.
DOI: https://doi.org/10.7554/eLife.49511.022

($N$ = 727 GP$_{STN}$ neurons in nine sections from three rats; *Figure 6A*). Triple immunofluorescence combined with a single retrograde tracer injection into the striatum was conducted to further elucidate the molecular identity of GP$_{CPu}$ neurons. The expression of Lhx6 (333/665) or FoxP2 (328/665) was again almost mutually exclusive. Only a small fraction (14.7%) of Lhx6-expressing neurons co-expressed PV (49/333) (*Figure 6B*). Therefore, GP$_{STN}$ neurons comprised PV(+) and/or Lhx6(+) prototypic neurons, whereas GP$_{CPu}$ neurons comprised arkypallidal neurons expressing FoxP2 and prototypic neurons expressing Lhx6 (*Fujiyama et al., 2016*; *Mallet et al., 2012*). Occasionally, double-retrogradely labeled neurons, namely bi-directionally projecting GP neurons (GP$_{Bi}$), which innervate both the STN and the striatum, were observed ($N$ = 113; *Figure 6A3*). Lhx6 was expressed in most GP$_{Bi}$ neurons (61/69), PV less frequently (19/72), and FoxP2 rarely (2/85), as expected. GP$_{Bi}$ neurons that expressed PV co-expressed Lhx6, although the number of neurons examined was small (5/6). These observations show that selective electrophysiological recordings from putative Lhx6(+) neurons can be conducted by targeting GP$_{Bi}$ neurons. We found that GP$_{Bi}$ neurons received cortical inputs (7/10 for M1 and 7/9 for M2; $N$ = 3 rats for each), although most of them exhibited oEPSCs with small amplitudes (*Figure 6C*). In fact, the frequency distributions of oEPSC amplitudes significantly differed between GP$_{Bi}$ and GP$_{CPu}$ neurons (*Figure 6D*; p=2.2 $\times$ 10$^{-16}$, Kolmogorov-Smirnov test). To clarify whether arkypallidal neurons are innervated by the motor cortex, the GP$_{CPu}$ neurons, in which oEPSCs were detected, were examined for immunofluorescence against FoxP2 ($N$ = 9). We found that eight out of nine GP$_{CPu}$ neurons expressed FoxP2 (*Figure 6E*), suggesting that arkypallidal neurons are preferentially innervated by the motor cortex. Again, although the comparison of oEPSC amplitudes among experiments is difficult to interpret, this could imply that arkypallidal neurons receive larger cortical inputs.

## Discussion

To summarize, we here report on the direct motor cortical innervation of GP neurons. Morphologically, the axon varicosity density in the GP reached 47% and 78% of that in the STN for M1 and M2 projections, respectively. Pallidostriatal neurons were more frequently innervated by the cortex than pallidosubthalamic neurons, and they often expressed FoxP2, indicating that they were arkypallidal neurons (see *Figure 7* for the schematic). To date, the fast excitation observed in the GP following cortical stimulation has been considered disynaptic excitation via the STN (*Nambu et al., 2000*). This does not fully contradict the present findings, since in a larger population of GP neurons, GP$_{STN}$ (or prototypic neurons), only 33% (from M1) and 52% (from M2) received motor cortical inputs. In addition, it is possible that extracellular unit recordings are biased towards neurons with relatively high firing frequencies, which include GP$_{STN}$ neurons (*Figure 5B*, *Table 2*). A question that remains is whether cortico-pallidal projections exist in primates including humans, as has been suggested (*Milardi et al., 2015*; *Smith and Wichmann, 2015*).

### Methodological consideration

In the present study, we employed tracer or viral vector injections to label cortico-pallidal projections in both morphological and electrophysiological experiments. Injections can extend to nearby areas, and some tracers can be transmitted both anterogradely and retrogradely, which results in the contamination of multiple projection pathways. This was however not the case in this study. First, as we show in *Figures 2* and *3*, cortical tracer injections into the motor area were restricted to single areas. Second, we also examined well characterized cortico-striatal and cortico-subthalamic projections (preliminary data can be found in *Karube et al., 2019*), and we found that the distributions of M1, M2, LO, and Cg axons differed in the striatum and STN, in line with earlier reports

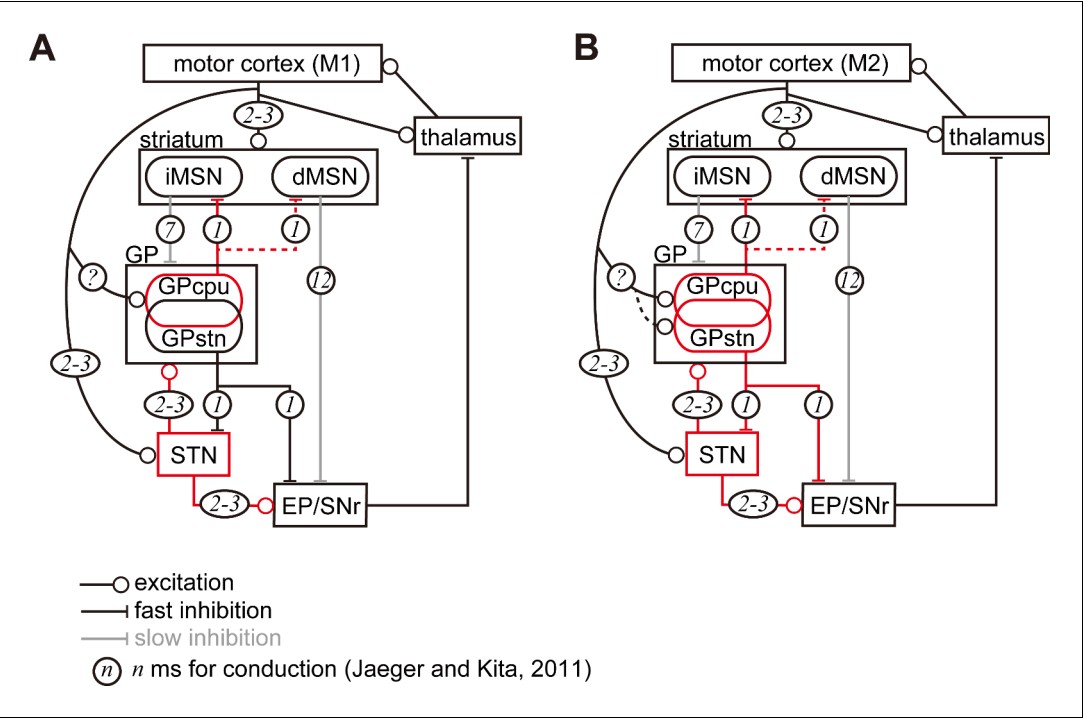

**Figure 7.** Diagram of cortico-basal ganglia-thalamus circuitry. The dotted lines represent the relatively weak innervation reported in this study for cortex to $GP_{STN}$, and for $GP_{CPu}$ to dMSNs (***Glajch et al., 2016***). The numbers in circles indicate estimated conduction times in milliseconds (ms) (***Jaeger and Kita, 2011***). (**A**) M1 activation induces excitation in the striatum, STN, and $GP_{CPu}$. Due to the electrophysiological properties of the neurons, STN and GP may be activated faster than striatal neurons. (**B**) M2 activation conveys additional excitation to the $GP_{STN}$ neurons. The possible information flow related to the present study is indicated by red lines. Note that inhibition from the GP is here considered faster (1 ms) than excitation from the STN (2–3 ms) (***Jaeger and Kita, 2011***). The actual timing of spike activity depends on neuron type, and excitation/inhibition interactions are complex.
DOI: https://doi.org/10.7554/eLife.49511.023

---

(***Afsharpour, 1985***; ***Alloway et al., 2010***; ***Brown et al., 1998***; ***Ebrahimi et al., 1992***; ***Hoover and Vertes, 2011***; ***Janssen et al., 2017***; ***Kincaid and Wilson, 1996***; ***Kolomiets et al., 2001***; ***Mailly et al., 2013***; ***Reep and Corwin, 1999***; ***Reep et al., 2008***). Third, we occasionally found some retrogradely labeled neurons in remote cortical areas, mainly in the temporal/insular area and posterior parietal area after injection into the motor areas. However, in approximately three-fourths of sections, no retrogradely labeled neuron was found. In the remaining sections, only less than eight pyramidal neurons per section were sparsely scattered. Thus, their contribution to axonal projections is likely to be negligible.

Another technical limitation that needs to be mentioned is the difficulty to precisely compare oEPSC amplitudes among experiments. As shown in *Figure 5D*, oEPSC amplitudes in the striatum and STN also varied among neurons. We cannot exclude the possibility that amplitude differences between $GP_{STN}$ and $GP_{CPu}$ or between M1-$GP_{CPu}$ and M2-$GP_{CPu}$ may be caused by inconsistencies in AAV infections, even though there were significant differences. We think that the potential variance in oEPSC amplitudes that stems from differences in the efficiency of viral vector infections is

**Table 2.** The maximum number of cortical boutons ($/100 \times 100 \ \mu m^2$) in the basal ganglia nuclei.

| Cortical area | GP | Striatum | STN |
|---|---|---|---|
| M1 (*N* = 4 rats) | 267, 200, 289, 141 | 911, 1300, 1464, 657, | 506, 432, 528, 408 |
| M2 (*N* = 4 rats) | 317, 737, 513, 92 | 1300, 2021, 1829, 398 | 420, 717, 637, 177 |

DOI: https://doi.org/10.7554/eLife.49511.019

evened out by the large number of GP neurons examined, but further research will be needed to come to a definite conclusion.

## Possible effects of cortical innervation in GP neurons

Cortico-pallidal projections were topographically organized; M1 and M2 axons were densely distributed in the CB(-) central part of the GP (*Figures 1* and *2*, *Figure 1—figure supplements 1* and *2*). Cg axons projected to the GP in a similar manner, but dense in the more medial GP. In contrast, LO provided fewer pallidal collaterals (*Figure 2—figure supplements 1* and *2*). Thus, each frontal cortical area likely has its own regions of interest in the GP. Our findings may explain the heterogeneous activity of GP neurons during movement (*Arkadir et al., 2004*; *DeLong, 1971*; *Dodson et al., 2015*; *Goldberg and Bergman, 2011*; *Mink and Thach, 1991a*; *Mink and Thach, 1991b*; *Turner and Anderson, 1997*). The firing phases of prototypic neurons (GP$_{STN}$) and arkypallidal neurons corresponding to FoxP2(+) GP$_{CPu}$ differ with regard to cortical activity (*Abdi et al., 2015*; *Mallet et al., 2012*; *Mallet et al., 2016*). This can be due to biased cortical innervation of the GP$_{CPu}$ (*Figure 5C, D, E, F*, *Figure 6C, E*). In addition, the above reports revealed that the firing phase was heterogeneous in prototypic neurons, too. We show here that one-third to half of all prototypic neurons received cortical inputs, which can contribute to their heterogeneity. The axon terminal density of cortico-pallidal projections was much sparser than that of cortico-striatal projections, but reached 50–80% of that of cortico-subthalamic projections. Considering that even in the connection between the STN and GP, which is one of the main synaptic connections in the indirect pathway, only 1–2% of all GP neurons converge onto a single STN neuron and vice versa (*Baufreton et al., 2009*; *Goldberg and Bergman, 2011*), sparse connections can still functionally work. Moreover, even if excitatory synaptic inputs did not reach action potential thresholds, the timing of subsequent action potentials can be altered (*Ermentrout, 1996*; *Schultheiss et al., 2010*). In addition, the expression of sodium channel characteristics of GP neurons can boost excitatory synaptic inputs (*Edgerton et al., 2010*; *Hanson et al., 2004*). Thus, the relatively sparse cortical innervation to the GP$_{STN}$ neurons could nevertheless affect a small but specific component of the basal ganglia circuitry.

Cortical innervation could affect the synchronization of GP neurons. During spontaneous firing, even most neighboring GP neuron pairs do not exhibit real correlation (*Bar-Gad et al., 2003*; *Goldberg and Bergman, 2011*; *Stanford, 2003*), even for pause periods (*Elias et al., 2007*). However, it has been reported that excitatory inputs may synchronize GP neurons effectively, as GP neurons tend to align with STN inputs rather than with striatal inhibitory inputs (*Goldberg et al., 2003*). Monosynaptic cortico-pallidal inputs, which are also excitatory, probably precede the inputs from the STN and can contribute to synchronization of GP neurons, which would strengthen STN-GP coupling (*Bevan et al., 2002a*; *Bevan et al., 2002b*; *Parent and Hazrati, 1995b*), thereby affecting the oscillatory phase and synchronization of the basal ganglia.

## Potential roles of the cortico-pallidal pathway in the cortex-basal ganglia-thalamus loop

We could not identify the origin of cortical neurons that provide axons in the GP. Pyramidal tract (PT) type pyramidal neurons are potential candidates, because some collaterals were emitted from thick axon trunks passing the GP (*Figure 1B*). The most recent finding in mice suggested that layers 5 and 6 cortical neurons project to the GP (*Abecassis et al., 2019*), although it is not clear whether they project only to the GP. PT neurons innervate multiple brain regions including the striatum and STN in rodents, however, their axon collaterals in the GP have not been reported yet (*Kita and Kita, 2012*; *Shepherd, 2013*; *Shibata et al., 2018*). The combinations of multiple cortical innervation onto the basal ganglia nuclei must influence information flow. Thus, it needs to be clarified whether the cortical neurons innervating the GP also innervate other basal ganglia nuclei.

To speculate potential effect of cortico-pallidal projections, we follow the related neural circuitries in the basal ganglia. Once M1 and/or M2 are activated, membrane properties of STN and GP neurons will contribute to earlier and more efficient excitation than those of striatal neurons (*Jaeger and Kita, 2011*; *Figure 7*). Since GP$_{CPu}$ neurons are activated by direct M1 inputs, they may increase the firing more significantly than GP$_{STN}$ neurons and be more in phase with cortical excitation. With regard to synaptic targets of GP$_{CPu}$ neurons, *Mallet et al. (2012)* reported that both

MSNs and interneurons in the striatum are innervated by arkypallidal neurons. In mice, Npas1(+) neurons, which overlap with Lhx6(+) and FoxP2(+) GP$_{CPu}$ neurons, synapse onto iMSNs and dMSNs. The average inhibitory postsynaptic current amplitude evoked by stimulation of axon terminals of Npas1(+) neurons is significantly larger in iMSNs than dMSNs (*Glajch et al., 2016*). Therefore, the inhibition of MSNs by GP$_{CPu}$ could counteract corticostriatal excitation, and weaken iMSN activity. Because iMSNs are likely to inhibit all GP neuron types equally (*Hernández et al., 2015*), decreased iMSN activity may reduce the inhibition of GP neurons, thereby augmenting the inhibition of STN, EP, and SNr. As the operation of this trisynaptic circuit should be slower than monosynaptic cortico-STN excitation (*Figure 7*), the M1 cortico-pallidal pathway may act as a delayed terminator of the hyperdirect pathway. GP$_{CPu}$ can also inhibit dMSNs, which in turn disinhibit the EP/SNr.

If iMSN activity is suppressed by the GP$_{CPu}$, what controls or suppresses GP$_{CPu}$ excitation? The most parsimonious explanation is that decreased cortico-pallidal activity can weaken GP excitation. In addition, it is noteworthy that striatal neurons receive contralateral cortical excitation as well as ipsilateral input (*Wilson, 1986*; *Wilson, 1987*), whereas the GP and STN receive only ipsilateral excitation (*Figure 1D*). Thus, it could be possible that the accumulating cortico-striatal excitation evoked by both hemispheres may overcome the pallido-striatal inhibition evoked by ipsilateral cortico-pallidal and cortico-STN excitation. If that was the case, then, the iMSNs can suppress GP neurons. It is also possible that axon collaterals of dMSNs, which would receive weaker inhibition from GP than iMSNs, inhibit the GP (*Fujiyama et al., 2011*; *Kawaguchi et al., 1990*; *Lévesque et al., 2003*; *Wu et al., 2000*). In addition, pallidostriatal inhibition of striatal interneurons can possibly disinhibit MSN activity. Finally, via local axons of GP neurons (*Fujiyama et al., 2016*; *Mallet et al., 2012*), mutual inhibition (*Bugaysen et al., 2013*; *Mastro et al., 2017*; *Sadek et al., 2007*) may work to terminate transient excitation of the GP.

In the case of M2 inputs to the GP, the GP$_{STN}$ pathway can also be activated, as well as the GP$_{CPu}$ pathway described above (*Figure 7B*). M2-GP$_{STN}$ circuitry may be more sensitive to timing, because the GP and STN form bidirectional connections, and the cortico-pallidal pathway can act as fast as the hyperdirect pathway. These two pathways may compete: if GP$_{STN}$ is activated first, it will suppress STN; if STN is activated first, GP will be excited directly by the cortex and via the STN, in turn inhibiting the STN. Taken together, it is implied that the net effect of M2-GP$_{STN}$ pathway activity is likely to be suppression of the STN. However, basal ganglia activity may be strongly affected by competition between M2-pallidal and hyperdirect pathways. Because the hyperdirect pathway contributes to the cessation of ongoing actions, the cortico-pallidal pathway may cancel this cessation signal. *Mallet et al. (2016)* proposed a two-step model for cancellation via cooperation between the STN and GP, especially arkypallidal neurons. Our current findings shed light on the linking of complex stop/cancel sequences. In addition, GP$_{STN}$ neurons innervate SNc dopaminergic neurons in a cell type specific manner (*Mastro et al., 2014*; *Oh et al., 2017*). Based on our results indicating that M2 but not M1 projects to the dorsolateral striosomes (*Figure 1—figure supplement 3*) and on the fact that striosomal dMSNs innervate the SNc (*Fujiyama et al., 2011*; *Gerfen, 1985*), M2-driven pathways may modulate dopamine signaling via the striatum and GP$_{STN}$. Recently, *Magno et al. (2019)* revealed that M2 activation can relieve the motor dysfunction of Parkinson's disease in mice. The neural circuitry proposed here is likely to contribute to such an effect.

The findings of recent in vivo experiments in rodents have revealed functions of M2. M2 contributes to the preparatory function (*Li et al., 2015*; *Svoboda and Li, 2018*), as observed in primates (*Wise, 1985*), or increases its influence on other cortical areas during the preparatory phase (*Makino et al., 2016*; *Makino et al., 2017*). GP neurons are also suggested to compute action selection (*Bogacz et al., 2016*; *Goldberg and Bergman, 2011*), which likely relates to the preparation of movement, and the M2-pallidal pathway may therefore contribute. It has been also suggested that M2 is involved in the integration of movement with sensory or internal information (*Barthas and Kwan, 2017*; *Saiki et al., 2014*), including posture coding (*Mimica et al., 2018*). It is possible that M2-GP innervation reflects sensory signal information and contributes to the fine tuning of ongoing movement such as the adaptation of chosen behaviors. Along with possibly involving the dopamine system, the M2-GP pathway may relate more to the plastic and integrated phases of movement, which may require sensitive control using two GP cell types. However, functional differentiation between M1 and M2 must be more complex, since concurrent and similar activities of M1 and M2 neurons have been reported, at least for certain movements (*Saiki et al., 2014*; *Soma et al., 2017*). Parallel connections between M1 and M2 also affect the aforementioned neural circuitry

(*Ueta et al., 2013*; *Ueta et al., 2014*). To conclude, the functional relevance and importance of cortico-pallidal projections from M1/M2 remain to be resolved in detail.

## Materials and methods

Animal experiments were approved and performed in accordance with the guidelines for the care and use of laboratory animals established by the Committee for Animal Care and Use (Permit Number: A16008, A17001, A18001, A19036) and the Committee for Recombinant DNA Study (Permit Number: D16008, D17001, D18001, D19036) of Doshisha University. All efforts were made to minimize animal suffering and the number of animals used.

### Animal surgery for injection of neural tracers and viral vectors

Wistar SLC rats (Japan SLC Inc, Hamamatsu, Japan; $N = 52$ of both sexes for electrophysiological experiments and $N = 16$ male rats for morphological experiments) were anesthetized with intramuscular injection of a mixture of ketamine (Ketalar; Daiichi-Sankyo, Tokyo, Japan; 40 mg/kg) and xylazine (Bayer HealthCare, Tokyo, Japan; 4 mg/kg). A small amount (0.05 mL) of the mixture was additionally injected every 15 min during any prolonged surgery (>1 hr). Body temperature was monitored and controlled at 37°C with the aid of a heating device (World Precision Instruments [WPI], Sarasota, FL, USA). Craniotomy was performed with a drill at an appropriate position on the skull based on the rat brain atlas (*Paxinos and Watson, 2007*). A glass pipette (tip diameter, 30–60 μm) was used for all injections. For anterograde tracers, biotinylated dextran amine (BDA, 10 kDa; 10% solution dissolved in PB; Thermo Fisher Scientific, Waltham, MA, USA) was injected using either a brief air pulse (10–20 psi for 40–100 ms) controlled with a coordinated valve system (PV 820, WPI) or using electrophoresis (0.5–2 μA of positive current, 7 s ON/7 s OFF for 80 cycles) with a current controller (WPI). PHA-L (Vector Laboratories, Burlingame, CA, USA; 2.5% dissolved in 10 mM $Na_2HPO_4$, pH 8.0) was injected using electrophoresis (1–5 μA positive current, 7 s ON/7 s OFF for 80 cycles). AAV vectors encoding fluorophores were injected using air pressure (1.5 × $10^{10}$ vg [vector genomes]/μL, *AAVdj-hSyn-hChR2(H134R)-mCherry*; the plasmid a gift from Karl Deisseroth; Addgene plasmid #26976; http://n2t.net/addgene:26976; RRID: Addgene_26976). For retrograde tracers, cholera toxin subunit B (CTB) conjugated with Alexa Fluor 488 or 555 (Thermo Fisher Scientific) and fluorophore-labeled beads (Green Retrobeads IX, LumaFluor Inc, Durham, NC, USA; FluoSphere Orange 0.04 μm, Thermo Fisher Scientific) were injected using air pressure. For labeling cortical neurons, injections were performed at two to three depths typically at 400 μm-intervals. The stereotaxic coordinates (*Paxinos and Watson, 2007*) of injections in the centers of each cortical area were as follows (the actual injection locations were normalized based on skull size using the distance between bregma and lambda): for rostral M1, 2.0 mm rostral from bregma (A/P r2.0) and 2.5 mm lateral from the midline (M/L 2.5), depth 0.5–1.2 mm from cortical surface; for caudal M1: A/P 0.0, M/L 2.6, depth 0.5–1.2; for M2: A/P r4.2, M/L 1.9, depth 0.5–1.2 at a 30° angle rostral from vertical; for lateral orbitofrontal area (LO): A/P r3.4, M/L 2.9, depth 4.0. Because the Cg is located at the medial surface of the frontal cortex and is enclosed by M2, contamination of M2 following vertical injections into the Cg would have been unavoidable. Therefore, for injections into the Cg, the contralateral medial frontal area was removed using an aspiration needle and vacuum pump to expose the medial surface of the frontal cortex of the targeted hemisphere. An injection electrode was then inserted into the Cg at an angle of 45° (A/P r1.8, M/L 0.5) and the contents of the electrode ejected into the Cg at two depths (0.5 and 1.0) using air pressure. Tracer injections into the striatum were performed to efficiently label GP neurons projecting to the dorsal striatum, pressure injection was applied in three tracks (two depths for each track) at A/P r2.0, M/L 2.5, depth 3.8; A/P r1.0, M/L 3.5, depth 4.2; and A/P 0.0, M/L 3.7, depth 4.0. After injections, the skin was sutured and 2.5 mg/kg of butorphanol (Vetorphale, Meiji Seika Pharma, Tokyo, Japan) was subcutaneously injected as an analgesic. The animals were allowed to recover before further experimentation. We allowed 2–4 d of survival time for BDA or retrograde tracers, 5–8 d for PHA-L, and 2–4 weeks for AAV. The age of the rats used for the morphological experiments ranged from 7 to 12 weeks.

### Immunohistochemistry

After the survival period, rats were deeply anesthetized with an intraperitoneal injection of sodium pentobarbital (100 mg/kg; Kyoritsu Seiyaku Corporation, Tokyo, Japan) and perfused with pre-

fixative (sucrose 8.5% w/v, MgCl$_2$ 5 mM; dissolved in 20 mM PB) followed by fixative (2 or 4% para-formaldehyde and 0.2% picric acid with or without 0.05% glutaraldehyde in 0.1 M PB, pH 7.4) through the cardiac artery. The brains were post-fixed in situ for 2–3 hr, and then removed and washed with PB several times. Sagittal or coronal sections 50 μm thick were cut using a vibratome (Leica VT1000, Leica Instruments, Wetzlar, Germany) or freezing microtome (Leica SM 2000R), and stored in PB containing 0.02% NaN$_3$ until further use.

For immunoreaction, sections were incubated with primary antibody diluted in incubation buffer consisting of 10% normal goat serum, 2% bovine serum albumin, and 0.5% Triton X in 0.05 M Tris buffered saline (TBS) overnight at room temperature (RT) or for 2–3 d at 4°C. For immunofluorescence, after rinsing with TBS three times, the sections were incubated with the secondary antibodies conjugated with fluorophores for 3 hr at RT. After three rinses, the sections were dried on glass slides and coverslipped with antifade mounting medium (ProLong Gold, Vector, Burlingame, CA). For brightfield specimens, the sections were incubated with a biotinylated secondary antibody followed by rinses and then reacted with ABC solution (1:200 dilution; Vector Elite) for 3 hr at RT, and visualized with 3,3′-diaminobenzidine (DAB) and/or Ni-DAB. The sections were dehydrated with a graded series of ethanol, delipidated with xylene, and finally embedded with M·X (Matsunami Glass Ind., Ltd., Osaka, Japan). The primary and secondary antibodies used in this study are listed in *Supplementary file 1* Tables 1 and 2.

## Image acquisition

Photomicrographs of brightfield specimens were captured using a CCD camera (DP-73, Olympus, Tokyo, JAPAN) equipped with a BX-53 microscope (Olympus) using 4× (numerical aperture [N.A.] 0.13), 10× (N.A. 0.3), 40× (N.A. 0.75), 60× (N.A. 0.9) or 100× (N.A. 1.4; oil-immersion) objectives. Photomicrographs were analyzed using Fiji (a distribution of Image J) (*Schindelin et al., 2012*), and Adobe photoshop (Adobe Systems Incorporated, Sam Jose, CA, USA). The brightness of digitized images was adjusted using the adjust-level function of these applications. To obtain a multifocus image, images were captured with 1 μm steps and processed with the 'extended depth of focus' Fiji plugin. Fluorescent images were captured using an Orca Spark CMOS camera (Hamamatsu photonics, Hamamatsu, Japan) or a DP-73 camera equipped with a BX-53 microscope. To quantify the molecular expression patterns of GP neuron types, immunofluorescent images were acquired using a confocal microscope (FV1200, Olympus) with 40× (N.A. 0.95) or 100× (N.A. 1.35; silicon oil immersion) objectives.

## In vitro slice recordings

Basal ganglia neurons were recorded using in vitro on cell and whole cell patch clamp. Rats of both sexes (N = 52; postnatal 30–65 d) were deeply anesthetized with isoflurane and perfused with 25 mL of ice-cold modified artificial cerebrospinal fluid (ACSF; N-methyl-D-glucamine, 93; KCl, 2.5; NaH$_2$PO$_4$, 1.2; NaHCO$_3$, 30; HEPES, 20; glucose, 25; sodium ascorbate, 5; thiourea, 2; sodium pyruvate, 3; MgCl$_2$, 10; and CaCl$_2$, 0.5; all in mM; pH was adjusted to 7.3 with HCl). All ACSFs were continuously aerated with 95/5% O$_2$/CO$_2$. Brains were removed and immersed in ice-cold modified ACSF for 2 min. Coronal slices 300 μm thick were cut using a vibratome (7000smz-2, Campden, Leicestershire, UK) and incubated with modified ACSF at 32°C for 15 min. The slices were transferred to normal ACSF (NaCl, 125; KCl, 2.5; CaCl$_2$, 2.4; MgCl$_2$, 1.2; NaHCO$_3$, 25; glucose, 15; NaH$_2$PO$_4$, 1.25; pyruvic acid, 2; lactic acid, 4; all in mM) at RT. After 1 hr of recovery, slices were moved into a recording chamber thermostatted at 30°C. A whole-cell glass pipette of 4–6 MΩ was filled with intracellular solution (K-gluconate, 130; KCl, 2; Na$_2$ATP, 3; NaGTP, 0.3; MgCl$_2$, 2; Na$_4$EGTA, 0.6; HEPES, 10; biocytin, 20.1; all in mM). The pH was adjusted to 7.3 with KOH, and the osmolality was adjusted to ~290 mOsm. Target brain regions were identified with the aid of a fluorescence microscope (BX-51WI, Olympus) using a × 40 water-immersed objective lens. Voltage and current clamp recordings were low-pass filtered at 10 kHz and recorded using EPC10 (HEKA Elektronik Dr. Schulze GmbH, Lambrecht/Pfalz, Germany) with a sampling rate of 20 kHz. Spontaneous firings were recorded in an on-cell mode for at least 2 min, then neurons were recorded in a whole-cell mode. The series resistance was examined by applying a brief voltage pulse of −10 mV for 10 ms and was confirmed to be less than 25 MΩ during recording. Shortly (less than 1 min) after achieving whole cell configuration, the firing responses to 1 s depolarizing current pulses (maximum intensity was 1000 pA, increasing

in 50 pA steps) were recorded in current clamp mode. Passive membrane properties were monitored as responses to 1 s hyperpolarizing current pulses.

For photoactivation of ChR2, a 470 nm light-emitting diode (LED; BLS-LCS-0470-50-22, Mightex Systems, Pleasanton, CA, USA) was used at full field illumination through a 40 × water immersion objective. Five-millisecond blue light pulses were applied at a maximum total power of ~4 mW, at which neuronal responses were saturated. One photo stimulation sequence was composed of 10 light pulses at 10 Hz, and repeated for 10 to 15 times with an interval of 1 s. In some experiments, low concentrations of TTX, 1 μM, and 4-amino pyridine (1 mM) were added to the ACSF to isolate monosynaptic currents (*Petreanu et al., 2009*; *Shu et al., 2007*). CNQX (10 μM) and AP5 (20 μM) were applied to inhibit glutamatergic synaptic currents (*N* = 10), and SR95531 (gabazine; 20 μM) was applied (*N* = 10) to prevent GABA$_A$ receptor-mediated synaptic currents. All pharmacological reagents were purchased from Tocris Bioscience (Bristol, UK).

Slices were then fixed with a mixture of 4% paraformaldehyde, 0.05% glutaraldehyde, and 0.2% picric acid in 0.1 M PB overnight at 4°C. Fixed slices were rinsed with PB (3 × 10 min). In some cases, they were then re-sectioned into 50 μm-thick sections. The sections were incubated with 1% H$_2$O$_2$ in 0.05 M TBS for 30 min at RT to deplete endogenous peroxidases, then rinsed with TBS three times. The sections were incubated with CF350-conjugated streptavidin for 2 hr (1:3000; Biotium, Inc, Fremont, CA) for fluorescent investigation of biocytin-filled neurons. For immunoreaction against FoxP2, the slices were treated using the CUBIC protocol for tissue clearing (*Susaki et al., 2015*). Briefly, the slices were incubated with 15% sucrose in PB for 3 hr, followed by 30% sucrose in PB for 3 hr for cryoprotection. After two rounds of freezing and thawing with dry ice, the slices were cleared with CUBIC-1 solution overnight, composed of 25% urea, 25% N,N,N′,N′-tetrakis-(2-hydroxypropyl)-ethylenediamine and 15% polyethylene glycol mono-p-isooctylphenyl ether in DW. After washing with PBS containing 0.3% Triton X (PBS-X), the slices were incubated with CF350 conjugated streptavidin overnight. After washing, the slices were incubated overnight with CUBIC-2 solution (10% 2,2′,2″-nitrilotriethanol, 50% sucrose, and 25% urea in DW). The cleared sections were investigated using confocal microscopy with a long working distance lens to capture the image of biocytin-filled neurons and its retrobeads labeling. Once recorded neurons were identified, the slices were washed with PBS-X and incubated with anti-FoxP2 antibody overnight. Due to the limited number of fluorescent channels, FoxP2-immunoreaction was detected using the same wavelength as fluorescence of one of the retrobeads. We confirmed that the fluorescent signals of retrobeads could be easily distinguished from those of FoxP2, with FoxP2 localized in the nucleus and retrobeads concentrated in the cytosol, and the signal size larger for FoxP2 than for retrobeads (*Figure 4E*). Usually, fluorescence of retrobeads is more or less diminished at the time of observation of FoxP2 immunofluorescence (*Figure 4E*). For brightfield microscopy, the sections were incubated with ABC (1:200) overnight at 4°C. Biocytin-filled neurons were visualized for light microscopy with Ni-DAB using H$_2$O$_2$ at a final concentration of 0.01%. The sections were dried on a glass slide and coverslipped with EcoMount (Biocare Medical, LLC, Concord, CA) or Mount-Quick (Daido Sangyo, Toda, Japan).

## Data analysis

### Quantification of cortico-pallidal axon density

Many thick cortical axon bundles with strong fluorescence passed through the GP, which largely disrupted proportional relationships between fluorescence intensity and axon varicosity (or bouton) density. Therefore, we manually counted axonal varicosities in each ROI using brightfield microscopic images. ROI size was 233 × 173 μm$^2$, which was enough to sample the region with densest axon terminals (*Figure 1—figure supplements 1* and *2*). Because cortical boutons were observed in the striatum and STN of the same animal, these were also counted, and the resultant bouton density was compared with that of the GP to decrease the potential effect of tracer injection variability.

### Analysis of electrophysiological data

Our analysis method has been previously described (*Mizutani et al., 2017*; *Oh et al., 2017*). Slice recording data were analyzed using Igor Pro 7 (Wave metrics Inc, Portland, OR) with the aid of the Neuromatic plugin (http://www.neuromatic.thinkrandom.com) (*Rothman and Silver, 2018*) and custom-built procedures. The input resistance was determined via the linear fitting of voltage responses

to hyperpolarized current pulses (−20 to −100 pA with 20 pA steps). The membrane time constant was calculated from the voltage response to a −50 pA current pulse. To identify oEPSCs, each trace was smoothed with a 0.2 ms moving time window (moving average of 4 consecutive recording points), and all traces were aligned with the onset of each photo stimulation to calculate the average trace. The baseline was defined as the mean current trace over a 50 ms period prior to each photo stimulation. the amplitude of the inward current with a stable delay after the onset of photo stimulation was measured from the baseline to the peak of the current. The inward current was identified as an oEPSC if the peak amplitude was 3-fold larger than the standard deviation of the baseline. The rising phase of the oEPSC was fitted linearly and extrapolated and the intersection point with the baseline was determined. The time of this intersection point was defined as the onset of the oEPSC. Latency was measured as the time from the onset of each photo stimulation to that of the oEPSC. Neurons in the striatum, basal nucleus of Meynert, and GP possessed distinct firing properties and were readily distinguished. In the GP, GABAergic projection neurons and putative cholinergic neurons were distinguishable by cell morphology and firing properties. Putative cholinergic neurons were excluded from the present data. The locations of recorded cells were visualized with biocytin and manually plotted on a digital image.

## Statistical comparisons

Averaged data are provided as mean ± standard deviation unless otherwise noted. Data comparison among more than two groups was performed using one-way ANOVA followed by post hoc Tukey tests, using R software (http://www.r-project.org/; R Project for Statistical Computing, Vienna, Austria). For small samples, the Kruskal-Wallis test was applied instead of ANOVA. Data comparison between two groups was performed using the Wilcoxon rank sum test. For comparison of proportion values, Fisher's exact test was used. To examine whether the mean of normalized values was significantly different from 1, a signed rank test for the mean was applied. For comparison of cumulative histograms, the Kolmogorov-Smirnov test was applied. Differences in data values were considered significant if $p < 0.05$. Significant differences are indicated using asterisks (*, $p < 0.05$; **, $p < 0.01$; ***, $p < 0.001$). All *p*-values are reported.

# Acknowledgements

The authors thank Dr. Yasuharu Hirai for helpful discussion and comments.

# Additional information

## Funding

| Funder | Grant reference number | Author |
| --- | --- | --- |
| Japan Society for the Promotion of Science | Grant-in-Aid for Scientific Research (C) 26350983 | Fuyuki Karube |
| Japan Society for the Promotion of Science | Grant-in-Aid for Scientific Research on Innovative Areas 16H01622 | Fuyuki Karube |
| Japan Society for the Promotion of Science | Grant-in-Aid for Scientific Research on Innovative Areas 16H06543 | Susumu Takahashi |
| Japan Society for the Promotion of Science | Grant-in-Aid for Scientific Research (B) 16H02840 | Susumu Takahashi |
| Japan Society for the Promotion of Science | Grant-in-Aid for Scientific Research (B) 16H03299 | Fumino Fujiyama |
| Japan Society for the Promotion of Science | Grant-in-Aid for Challenging Exploratory Research 15K12770 | Fumino Fujiyama |
| Japan Society for the Promotion of Science | Scientific Researches on Innovative Areas 26112006 | Fumino Fujiyama |

| Japan Society for the Promotion of Science | Grant-in-Aid for Scientific Research (A) 19H01131 | Susumu Takahashi |
| --- | --- | --- |

The funders had no role in study design, data collection and interpretation, or the decision to submit the work for publication.

## Author contributions

Fuyuki Karube, Conceptualization, Resources, Data curation, Software, Formal analysis, Funding acquisition, Validation, Investigation, Visualization, Methodology, Writing—original draft, Project administration, Writing—review and editing; Susumu Takahashi, Fumino Fujiyama, Resources, Funding acquisition, Writing—original draft; Kenta Kobayashi, Resources

## Author ORCIDs

Fuyuki Karube (iD) https://orcid.org/0000-0002-5365-3297

## Ethics

Animal experimentation: Animal experiments were approved and performed in accordance with the guidelines for the care and use of laboratory animals established by the Committee for Animal Care (Permit Number: A16008, A17001, A18001, A19036) and Use and the Committee for Recombinant DNA Study (Permit Number: D16008, D17001, D18001, D19036) of Doshisha University. All efforts were made to minimize animal suffering and the number of animals used.

## Decision letter and Author response

Decision letter https://doi.org/10.7554/eLife.49511.027
Author response https://doi.org/10.7554/eLife.49511.028

# Additional files

## Supplementary files

• Supplementary file 1. Primary and secondary antibodies used in this study.
DOI: https://doi.org/10.7554/eLife.49511.024

• Transparent reporting form DOI: https://doi.org/10.7554/eLife.49511.025

## Data availability

All data generated or analysed during this study are included in the manuscript and supporting files. Source data files in a Microsoft Excel format are provided for Table 2, for Figures 2C, 2D, 2E, 4A2, 4B2, 4B3, 4B4, 4C, 5B, 5C, 5D, 5F, 6A, 6B, 6C, 6D, and also for Figure 1-Figure supplement 2E, and Figure 2-Figure supplement 1D.

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
