## [Decision Letter]

**Acceptance summary:**

This study found a significant direct projection from the cortex to the globus pallidus (GP), an overlooked input from the cortex to the basal ganglia circuit. This projection is quantitatively comparable to the cortical projection to the subthalamic nucleus (hyperdirect pathway). The authors have also found that this projection targets GP in a cell type specific manner. These findings prompt a revision of a commonly held view that the striatum and subthalamic nucleus are the predominant input structures of the cortico-basal ganglia circuit.

**Decision letter after peer review:**

Thank you for submitting your article "Motor cortex can directly drive the globus pallidal neurons in a projection neuron type dependent manner in rat" for consideration by *eLife*. Your article has been reviewed by three peer reviewers, and the evaluation has been overseen by a Reviewing Editor and Floris de Lange as the Senior Editor. The reviewers have opted to remain anonymous.

The reviewers have discussed the reviews with one another and the Reviewing Editor has drafted this decision to help you prepare a revised submission.

Summary:

The striatum has been often regarded as the sole input structure of the basal ganglia. This study shows that the premotor and motor cortices (M1 and M2, respectively) directly project to the globus pallidus. Although a previous study (Naito and Kita, 1994) reported this connection, this observation has been largely neglected in the field. The present study re-examines this connection using anterograde anatomical tracing and channelrhodopsin-2-assisted circuit mapping in slice, and concludes that the cortico-pallidal projection is as strong as the hyperdirect pathway (cortico-subthalamic projection).

All the reviewers thought that this is a potentially very important finding. However, the reviewers raised substantive concerns regarding the clarity of the manuscript and some technical issues. As you will see below, the original comments by the reviewers somewhat varied in terms of the points raised and enthusiasm. After discussions, however, we were able to narrow the essential points that the authors should address. We believe that the authors should be able to address these essential points with the current data although the reviewers pointed out various experiments that can improve the quality of the work. Although these additional experiments are optional following *eLife*'s policy, please consider these experiments as much as you can or provide use with your responses.

Essential revisions:

1) The reviewers thought that the current manuscript is difficult to read because it contains many results that are not directly related to the main story and that are, at least in part, reported in previous studies (reviewer 1, the point starting with "Finally, the paper jumps around quite a bit" and reviewer 3, point #1).

a) Significance and novelty of Figure 5 were not very clear. Please explain these more clearly.

b) Please remove Figure 6 (M1, M2 projections to the striatum), Figure 7 (distribution of STN-projecting cortical neurons), and Figure 8 (spatial distribution of cortico-STN axons).

2) The authors conclude that "the cortical excitation in the GP was as strong as that in the STN" but the evidence is not very convincing.

a) Please add a density analysis using wider regions of interest (ROIs) that encompass all of the GP and STN (reviewer 1, the point starting with "First, with the regard to").

b) Please interpret the electrophysiology results comparing inputs from M1 and M2 more quantitatively (reviewer 3, point 3).

c) (optional) For comparison, please perform electrophysiological characterization of cortico-STN connection (reviewer 1, the point starting with "The experiments in Figure 2 are meant to show").

3) The authors speculate that pyramidal tract (PT) neurons appear to be the source of input to GP neurons. It is important to know whether this is true or whether it is a different (new) population.

a) Retrograde labeling might clarify which neurons in the cortex are projecting to the GP (reviewer 1, the point starting with "There is a missing, obvious anatomical experiment"). During discussion, the reviewers thought that knowing this would increase the significance of this work. However, it is also pointed out that retrograde tracing may not be very straightforward with a conventional tracer due to passing fibers etc. and may require a different method such as rabies virus-mediated tracing. Due to these complexities, we thought that retrograde tracing could be optional. However, if you choose to do so, please reduce the emphasis on PT and discuss this point more carefully.

4) The authors suggest that "arkypallidal neurons, which do not send axons to the STN, could be a principle target of cortico-pallidal innervation". Reviewer 3 raised this issue but during discussion, it was discussed that FoxP2 is a nuclear protein and it might be difficult to detect in the proposed experiment. Thus, an EM investigation could be combined with retrograde labeling from the striatum (CPU). Although we thought this could be useful, it was also pointed out that the results do not advance beyond what the authors have done with slice electrophysiology. Please explain your view on this in the rebuttal letter.

The details of the above issues in the original review comments can be found below. Please use the essential points above when reading the original comments because the tone and the suggested experiments and analyses have been changed in some points after our discussions.

Reviewer #1:

The goal of this paper is characterize the direct projection between the cortex and the globus pallidus, and to place that projection within the broader context of cortico-basal ganglia circuitry. The authors conclude that the cortico-pallidal projection, at least from areas M1 and M2 in the rat, is just as strong as the hyperdirect pathway (cortico-STN projection). To my knowledge, no lab has replicated the important work of Naito and Kita demonstrating the cortico-pallidal projection, so this is a significant claim. My impression has been that this projection is largely ignored in studies of the basal ganglia, perhaps because people believe that it is too sparse to be very significant. Therefore, if it is, in fact, a projection equal to the one to the STN, the paper would be quite significant for our understanding of cortico-basal ganglia circuitry.

However, with what the authors are showing, I just don't think the case is strong enough yet. I think substantial revisions could help, but this involves collecting a moderate amount of additional data, and may be beyond the scope of *eLife*'s revisions policy.

First, with the regard to the data shown in Figure 1, I'm not sure that counting axon varicosity density within a small ROI at the point of densest projection is the best way to assess impact. These are small enough structures, I think the authors do need to count across the entire GP and STN (sampling stereologically, so it's not as great a burden). Isn't it possible that the projection to the STN is much wider-spread, so the overall impact is greater, but at its densest point, it will not have more varicosities? Also, it looks as though the "densest ROI" was chosen before counting, so couldn't this have been conflated across regions by passing axons, since the eye could not distinguish? What if varicosities are actually denser elsewhere? Finally, experimenter bias, however unconscious, may be a concern here based on how the ROI was selected.

Figure 1E1 is also the basis for the authors to conclude that the cortico-pallidal projection is just as strong as the cortico-STN projection. But with a p-value of 0.16, and only 3 subjects per injection, I have to wonder whether this analysis was drastically underpowered.

There is a missing, obvious anatomical experiment here that could substantiate their claims, and clarify the circuit The authors claim that these are likely collaterals off of PT neurons. They refer to the pallidal projections as collaterals throughout. I think this makes sense, and seems like the most logical circuit, but there aren't actually data to show that this is the case. They should do some retrograde injections in the GP to show the layer these cells are coming from, and to understand the density. They could compare to density with STN projections. Then some double injections in the GP and striatum to show that cortico-pallidal projections are also cortico-striatal projections. For all we know, at this point, this could be an entirely new population of projecting cells in M1/M2, not PT neurons at all!

The experiments in Figure 2 are meant to show that the cortico-pallidal pathway is a functional connection. But it seems like there's a big missing piece of the puzzle, and that's what that figure looks like for the cortico-STN projection. If the metrics look similar for the hyperdirect and cortico-pallidal pathways, it greatly bolsters the authors' argument. Hyperdirect experiments would also act as an internal control: I have some concern that the addition of the ChR2 is inflating the strength of the connection. And how should we judge the measures of strength shown? Is this strong? Is it weak?

Finally, the paper jumps around quite a bit, and there are several experiments, namely those from Figures 6-9, as well as half of Figure 5, that just don't seem to belong here. They don't relate the central point of the cortico-pallidal circuit, and just provide minor replications of prior work. I recommend the authors remove them entirely, and perhaps start building a second paper about circuit topographies. The current paper should instead be filled out with the revisions described above.

- The experiments shown in Figure 5A-B seem out of place. The authors are showing the proportion of GP-str and GP-stn cells that also label with PV, Lhx6, and FoxP2, but there is no reference to whether these GP cells are receiving direct cortical input. It doesn't fit with the other experiments.

- The experiments shown in Figure 6 also do not belong. They simply replicate abundant prior work showing the relationship between M1/M2 projections and striosome/matrix (Donoghue and Herkenham, 1986; Gerfen, 1989; Ebrahimi et al., 1992; and so on).

- Similarly, Figure 7, which demonstrates the topography of the hyperdirect pathway, is not strongly related to the goals of the paper. And the experiments have been done before (Janssen et al., 2010; Canteras et al., 1990; Kitai and Denieu, 1981; and so on). Instead, the figures showing the LO-pallidal and CG-pallidal projections should be pulled out of the supplement and put in the main text.

- Same with Figures 8 and 9.

Reviewer #2:

The manuscript submitted by Karube et al., characterized the cortical innervation principally coming from the primary (M1), the secondary (M2) motor cortex to the basal ganglia, in particular, the globus pallidus (GP) and the subthalamic nucleus (STN). The authors also looked at the projection from the orbitofrontal (OF), and the cingulate cortex (Cg) to the GP, the STN, and the striatum. The cortico-pallidal projection has been known for some time but poorly characterized. Here the authors show that the cortical inputs principally excite the striatum-projecting GP neurons (GP-CPU) over the STN-projecting neurons (GP-STN). They also demonstrate that the GP-CPU neurons can be further subdivided into FoxP2+ (i.e. the arkypallidal neurons) and Lhx6+ cells (that represent the main population of GP neurons, the so-called prototypic cells) and that both these populations receive functional inputs from the cortex. The authors also characterize some interesting topographical organization between the different cortical areas and the GP or the STN. Overall this is a nice and interesting story that add novel findings highlighting the anatomical and functional complexity of these basal ganglia circuits. These findings also challenge the classic view that cortical inputs are exclusively integrated by the striatum and the subthalamic nucleus in the basal ganglia. Indeed, here the authors show that the density of varicosity in STN is similar to the one in GP.

I found that the only small/moderate limitation of the work is that the origin and cellular identity of this cortico-pallidal inputs could have been better characterized. Indeed, the authors mention (subsection “Motor cortex innervates the GP”, second paragraph) that the GP projections are exclusively ipsilateral implying that the cortical neurons providing these inputs are pyramidal tract (PT) neurons. However, previous work that has investigated the projection sites of PT neurons have not seen any collaterals to GP (Kita and Kita, 2012). What could explain these differences? It would have been nice to clarify this issue in the paper. Interestingly, two population of PT neurons have been identified based on their projection site (i.e. thalamus or medulla projecting, see Economo et al., 2018 DIO: 10.1038/s41586-018-0642-9) and it is possible that this PT neuronal heterogeneity is at the origin of a separate pathway to the GP.

Reviewer #3:

This is an interesting paper that addresses an important issue that can have a significant impact on our current view of information processing through the basal ganglia circuits in normal and diseased states. The main finding of this study is evidence for the existence of a direct monosynaptic cortico-pallidal projection that innervates preferentially arkypallidal neurons in the rat globus pallidus (GP). Because the GP is not recognized as a direct target of cortical afferents in current schemes of the basal ganglia circuitry, the existence of this connection will generate significant interest in the field and may open up new critical thoughts about substrates of basal ganglia function and dysfunction. However, the manuscript in its present form suffers of major organizational problems and need additional data to support more strongly the authors' conclusions. Some of the substantive concerns that must be addressed include:

1) In its present form, the paper is very hard to read and the main message (i.e. existence of the cortico-pallidal inputs to arkypallidal neurons) is lost into a series of other anatomical data that have some value, but not for this paper. I suggest that the authors significantly streamline their paper to make it focused solely on the cortico-pallidal projection.

2) The evidence for a monosynaptic M1/M2 inputs to arkypallidal neurons should be supported by electron microscopic studies of synaptic connections between M1/M2 terminals and arkypallidal neurons. The addition of solid EM data to this study would strengthen significantly the authors' conclusions.

3) The results of optogenetic studies shown in Figure 2 and 4 are hard to interpret because the amplitude of EPSC evoked in the different populations of GP neurons and the number of responding neurons to M1 or M2 terminals is highly dependent on the number of cortical terminals that express ChR2 in each area of light stimulation. Because the number of transfected terminals in the GP is variable between animals due to inherent technical issues related to the characteristics of the injection sites, extent of uptake of the virus by cortical neurons and efficiency of anterograde transport along cortico-pallidal axons, quantitative data that compare M1- and M2-mediated effects on GP neurons are difficult to interpret. A more rigorous analysis that takes into consideration these technical issues must be done to validate these data.

---

## [Author Response]

Essential revisions:1) The reviewers thought that the current manuscript is difficult to read because it contains many results that are not directly related to the main story and that are, at least in part, reported in previous studies (reviewer 1, the point starting with "Finally, the paper jumps around quite a bit" and reviewer 3, point #1).a) Significance and novelty of Figure 5 were not very clear. Please explain these more clearly.

This point deeply relates to the point 4. Please refer below. We modified the corresponding text to explain the point (subsection “M1 and M2 more frequently innervate GP neurons projecting to the striatum than those projecting to STN”).

b) Please remove Figure 6 (M1, M2 projections to the striatum), Figure 7 (distribution of STN-projecting cortical neurons), and Figure 8 (spatial distribution of cortico-STN axons).

We omitted the corresponding figures and texts. One of the reasons why we had added those figures is confirmation of our injection quality. As the reviewer 1 commented, our data for cortico-striatal and cortico-STN projection patterns are similar with previous qualitative investigations. We appreciated if you keep it in mind (since the reviewer unfairly suspected our morphological data on GP, irrespective of his acceptance of cortico-striatal and cortico-STN data derived from the same samples as cortico-GP data are obtained). Anyway, they were gone in the revision.

2) The authors conclude that "the cortical excitation in the GP was as strong as that in the STN" but the evidence is not very convincing.a) Please add a density analysis using wider regions of interest (ROIs) that encompass all of the GP and STN (reviewer 1, the point starting with "First, with the regard to").

As reviewer 1 pointed out, the location of ROI was not clearly shown, which lead difficulty for understanding. Thus, we added the images with low magnification to represent ROI location and topographical distribution of cortical axons in the STN and GP as shown in Figure 1—figure supplement 1 and 2, which can support the reasonability of the ROI size, 233 µm x 173 µm. In addition, we counted another sample for both M1 and M2 projections (now N = 4 in the revision; Figure 2C, D, E). Also, our presentation in the previous Figure 1—figure supplement 3 maybe not adequate. We counted additional sections along with the M/L dimension, and modified the figure as shown in Figure 2C, D.

We argued against the scientific importance of counting boutons in the entire area of the GP and STN. First, as mentioned in Materials and methods and Figure 1—figure supplement 1 and 2, the ROI size is enough large to sample the densest cortical projection field in the GP and especially in the STN, since, as we showed, those projections are highly topographic. Extending ROI further means counting the bouton number in less innervated regions, and we considered it is not important. Second, bouton count is highly time consuming, because the size of boutons is close to light microscope resolution which prevents reliable segmentation by any algorithms (hopefully AI will help in the near future), and bouton count has to be done completely manually.

b) Please interpret the electrophysiology results comparing inputs from M1 and M2 more quantitatively (reviewer 3, point 3).

This is an actually critical point. We added “methodological consideration” in Discussion, and state that comparing the oEPSC amplitude is difficult and not fully conclusive (subsection “Cortico-GP terminals elicit monosynaptic EPSCs”, last paragraph; subsection “M1 and M2 more frequently innervate GP neurons projecting to the striatum than those projecting to STN, last paragraph and subsection “Methodological consideration”, last paragraph).

c) (optional) For comparison, please perform electrophysiological characterization of cortico-STN connection (reviewer 1, the point starting with "The experiments in Figure 2 are meant to show").

We add comparison of oEPSC amplitude between STN and GP neurons derived from the same animal, the data are represented in Figure 4C and subsection “Cortico-GP terminals elicit monosynaptic EPSCs”, last paragraph. By the signed rank test for the mean, normalized oEPSC amplitude in GP neurons are not significantly different from 1, namely, the mean amplitude of STN neurons. We had already showed comparison of oEPSC time course in the previous Figure 3. However, as shown in the previous Figure 4 (Figure 5 in the revision), oEPSC amplitudes in STN neurons themselves largely varied from cell to cell. Thus, I wonder the editors and other reviewers could accept the above comparison (if not, we will omit it).

3) The authors speculate that pyramidal tract (PT) neurons appear to be the source of input to GP neurons. It is important to know whether this is true or whether it is a different (new) population.a) Retrograde labeling might clarify which neurons in the cortex are projecting to the GP (reviewer 1, the point starting with "There is a missing, obvious anatomical experiment"). During discussion, the reviewers thought that knowing this would increase the significance of this work. However, it is also pointed out that retrograde tracing may not be very straightforward with a conventional tracer due to passing fibers etc. and may require a different method such as rabies virus-mediated tracing. Due to these complexities, we thought that retrograde tracing could be optional. However, if you choose to do so, please reduce the emphasis on PT and discuss this point more carefully.

This is an important issue, and we admit lack of enough evidences. We added images and drawings of partial reconstruction of cortico-GP projections, which represent axon branches issue from thick main axons (Figure 1B; subsection “Motor cortex innervates the GP”, first paragraph). However, still it is not clear whether single pyramidal cells emit all collaterals in the striatum, GP, and STN. Therefore, as the above comment, we carefully discussed on the point (subsection “Potential roles of the cortico-pallidal pathway in the cortex-basal ganglia-thalamus loop”, second paragraph).

We examined AAV-retro injection into GP and STN, or into GP and pontine nucleus. We will prepare a new paper, in which the data and data removed from the revision will be included.

4) The authors suggest that "arkypallidal neurons, which do not send axons to the STN, could be a principle target of cortico-pallidal innervation". Reviewer 3 raised this issue but during discussion, it was discussed that FoxP2 is a nuclear protein and it might be difficult to detect in the proposed experiment. Thus, an EM investigation could be combined with retrograde labeling from the striatum (CPU). Although we thought this could be useful, it was also pointed out that the results do not advance beyond what the authors have done with slice electrophysiology. Please explain your view on this in the rebuttal letter.The details of the above issues in the original review comments can be found below. Please use the essential points above when reading the original comments because the tone and the suggested experiments and analyses have been changed in some points after our discussions.

Thanks to the comment, now we successfully detected immunoreaction for FoxP2 in recorded neurons using CUBIC, a tissue clearing method. Thus, we clarified arkypallidal neurons actually receive cortical inputs. We added the images in Figure 6E and modified the text (subsection “M1 and M2 more frequently innervate GP neurons projecting to the striatum than those projecting to STN”, last paragraph).

More difficulty we face is immunoreaction of Lhx6, which is occasionally unstable after slice recording. That is the reason why we did experiments shown in the previous Figure 5 (Figure 6 in the revision), to identify putative Lhx6 neurons without immunoreaction. We modified the corresponding text to explain the point (see the last two paragraphs of the aforementioned subsection).

Reviewer #1:[…] First, with the regard to the data shown in Figure 1, I'm not sure that counting axon varicosity density within a small ROI at the point of densest projection is the best way to assess impact. These are small enough structures, I think the authors do need to count across the entire GP and STN (sampling stereologically, so it's not as great a burden). Isn't it possible that the projection to the STN is much wider-spread, so the overall impact is greater, but at its densest point, it will not have more varicosities? Also, it looks as though the "densest ROI" was chosen before counting, so couldn't this have been conflated across regions by passing axons, since the eye could not distinguish? What if varicosities are actually denser elsewhere? Finally, experimenter bias, however unconscious, may be a concern here based on how the ROI was selected.Figure 1E1 is also the basis for the authors to conclude that the cortico-pallidal projection is just as strong as the cortico-STN projection. But with a p-value of 0.16, and only 3 subjects per injection, I have to wonder whether this analysis was drastically underpowered.

We added samples and added the sections for counting, and re-formulate Figure 1, 2, Figure 1—figure supplement 1, 2.

There is a missing, obvious anatomical experiment here that could substantiate their claims, and clarify the circuit The authors claim that these are likely collaterals off of PT neurons. They refer to the pallidal projections as collaterals throughout. I think this makes sense, and seems like the most logical circuit, but there aren't actually data to show that this is the case. They should do some retrograde injections in the GP to show the layer these cells are coming from, and to understand the density. They could compare to density with STN projections. Then some double injections in the GP and striatum to show that cortico-pallidal projections are also cortico-striatal projections. For all we know, at this point, this could be an entirely new population of projecting cells in M1/M2, not PT neurons at all!

We thank the reviewer for this important comment. We had conducted double retrograde tracer injections, but cannot exclude the possibility of labeling from passing fibers in the GP. We carefully described this point in the revision (subsection “Motor cortex innervates the GP”, first paragraph; subsection “Potential roles of the cortico-pallidal pathway in the cortex-basal ganglia-thalamus loop”, second paragraph). We had started double retrograde labeling using AAV-retro and/or rabies viral vectors, which will be a piece of a new paper.

The experiments in Figure 2 are meant to show that the cortico-pallidal pathway is a functional connection. But it seems like there's a big missing piece of the puzzle, and that's what that figure looks like for the cortico-STN projection. If the metrics look similar for the hyperdirect and cortico-pallidal pathways, it greatly bolsters the authors' argument. Hyperdirect experiments would also act as an internal control: I have some concern that the addition of the ChR2 is inflating the strength of the connection. And how should we judge the measures of strength shown? Is this strong? Is it weak?

We had already shown the comparison of cortico-GP and cortico-STN EPSCs in the previous Figure 3, 4 (Figure 4, 5 in the revision, respectively), as the comment, we intended to employ them as internal control, since cortico-striatal and cortico-STN synapses are broadly accepted as functional.

Honestly saying, we cannot understand why “that's what that figure looks like for the cortico-STN projection“, since no critical reason was raised. We added the comparison of EPSC amplitude between GP and STN (Figure 4C and subsection “Cortico-GP terminals elicit monosynaptic EPSCs”, last paragraph). By the signed rank test for the mean, normalized oEPSC amplitude in GP neurons are not significantly different from 1, namely, the mean amplitude of STN neurons.

Finally, the paper jumps around quite a bit, and there are several experiments, namely those from Figures 6-9, as well as half of Figure 5, that just don't seem to belong here. They don't relate the central point of the cortico-pallidal circuit, and just provide minor replications of prior work. I recommend the authors remove them entirely, and perhaps start building a second paper about circuit topographies. The current paper should instead be filled out with the revisions described above.

We had planned to place our findings in the context of whole basal ganglia pathways, also these data could support our morphological analysis quality, since cortico-striatal projections have been widely examined, and we got similar results (although the quantitative data we had shown is not reported so far). However, the comment is reasonable, and we excluded Figure 6-9 from the current manuscript.

- The experiments shown in Figure 5A-B seem out of place. The authors are showing the proportion of GP-str and GP-stn cells that also label with PV, Lhx6, and FoxP2, but there is no reference to whether these GP cells are receiving direct cortical input. It doesn't fit with the other experiments.

We thank the comment, and our description on the point was not enough. We think these data are necessary to study GP neuron types which receives cortical inputs. Since immunohistochemistry for Lhx6 after in vitro slice recordings were unstable, we cannot directly examine that GP neurons receiving cortical inputs frequently express either Lhx6 or FoxP2. Therefore, we showed first double retrograde labeling cells, which project both STN and the striatum, express highly preferentially Lhx6. It means if we targeted bi-directionally projection cells, they are likely to be Lhx6 neurons. Certainly, as the comment, it is not direct evidence, however, we think at least it is scientifically logical. We modified the corresponding description (subsection “M1 and M2 more frequently innervate GP neurons projecting to the striatum than those projecting to STN”, last two paragraphs), and add the new immunofluorescence experiment to detect FoxP2 in the recorded neurons (Figure 6E; subsection “M1 and M2 more frequently innervate GP neurons projecting to the striatum than those projecting to STN”, last paragraph).

- The experiments shown in Figure 6 also do not belong. They simply replicate abundant prior work showing the relationship between M1/M2 projections and striosome/matrix (Donoghue and Herkenham, 1986; Gerfen, 1989; Ebrahimi et al., 1992; and so on).- Similarly, Figure 7, which demonstrates the topography of the hyperdirect pathway, is not strongly related to the goals of the paper. And the experiments have been done before (Janssen et al., 2010; Canteras et al., 1990; Kitai and Denieu, 1981; and so on). Instead, the figures showing the LO-pallidal and CG-pallidal projections should be pulled out of the supplement and put in the main text.- Same with Figures 8 and 9

The above papers did not strictly show the differential axonal distribution between M1 and M2, and quantitative comparison is missing. Anyway, we excluded the corresponding description in the revision.

Reviewer #2:[…] I found that the only small/moderate limitation of the work is that the origin and cellular identity of this cortico-pallidal inputs could have been better characterized. Indeed, the authors mention (subsection “Motor cortex innervates the GP”, second paragraph) that the GP projections are exclusively ipsilateral implying that the cortical neurons providing these inputs are pyramidal tract (PT) neurons. However, previous work that has investigated the projection sites of PT neurons have not seen any collaterals to GP (Kita and Kita, 2012). What could explain these differences? It would have been nice to clarify this issue in the paper. Interestingly, two population of PT neurons have been identified based on their projection site (i.e. thalamus or medulla projecting, see Economo et al., 2018 DIO: 10.1038/s41586-018-0642-9) and it is possible that this PT neuronal heterogeneity is at the origin of a separate pathway to the GP.

We appreciate the critical comment. This is one of the most interesting points, and our current data are not enough. Thus, we carefully stated the neuronal origin of this innervation (subsection “Potential roles of the cortico-pallidal pathway in the cortex-basal ganglia-thalamus loop”, second paragraph). We have started to label them using retrograde viral vectors for the next publication.

Reviewer #3:[…] The manuscript in its present form suffers of major organizational problems and need additional data to support more strongly the authors' conclusions. Some of the substantive concerns that must be addressed include:1) In its present form, the paper is very hard to read and the main message (i.e. existence of the cortico-pallidal inputs to arkypallidal neurons) is lost into a series of other anatomical data that have some value, but not for this paper. I suggest that the authors significantly streamline their paper to make it focused solely on the cortico-pallidal projection.

Thanks the comment. We divided the previous version and focused on cortical projection in the revision.

2) The evidence for a monosynaptic M1/M2 inputs to arkypallidal neurons should be supported by electron microscopic studies of synaptic connections between M1/M2 terminals and arkypallidal neurons. The addition of solid EM data to this study would strengthen significantly the authors' conclusions.

This is an actually important point and we can successfully detected FoxP2 immunoreaction after slice recording with the aid of CUBIC method. Eight of 9 examined GP_CPu_ neurons in which EPSC was observed expressed FoxP2. We added the images as Figure 6E, and corresponding texts (subsection “M1 and M2 more frequently innervate GP neurons projecting to the striatum than those projecting to STN”, last paragraph).

3) The results of optogenetic studies shown in Figure 2 and 4 are hard to interpret because the amplitude of EPSC evoked in the different populations of GP neurons and the number of responding neurons to M1 or M2 terminals is highly dependent on the number of cortical terminals that express ChR2 in each area of light stimulation. Because the number of transfected terminals in the GP is variable between animals due to inherent technical issues related to the characteristics of the injection sites, extent of uptake of the virus by cortical neurons and efficiency of anterograde transport along cortico-pallidal axons, quantitative data that compare M1- and M2-mediated effects on GP neurons are difficult to interpret. A more rigorous analysis that takes into consideration these technical issues must be done to validate these data.

The comment is critical and we agree with it, and our description was too careless. We tried to apply some normalization, but we cannot overcome disadvantages. In the revision, we modified the text to emphasize technical limitation, and intend to interpret the results more carefully (subsection “Cortico-GP terminals elicit monosynaptic EPSCs”, last paragraph; subsection “M1 and M2 more frequently innervate GP neurons projecting to the striatum than those projecting to STN, last paragraph and subsection “Methodological consideration”, last paragraph).